# The *Ustilago maydis* repetitive effector Rsp3 blocks the antifungal activity of mannose-binding maize proteins

Lay-Sun Ma[1], Lei Wang[1,5], Christine Trippel[1,6], Artemio Mendoza-Mendoza[1,7], Steffen Ullmann[1,8], Marino Moretti[1], Alexander Carsten[1], Jörg Kahnt[2], Stefanie Reissmann[1], Bernd Zechmann[3], Gert Bange[4] & Regine Kahmann[1]

To cause disease in maize, the biotrophic fungus *Ustilago maydis* secretes a large arsenal of effector proteins. Here, we functionally characterize the repetitive effector Rsp3 (<u>r</u>epetitive <u>s</u>ecreted <u>p</u>rotein 3), which shows length polymorphisms in field isolates and is highly expressed during biotrophic stages. Rsp3 is required for virulence and anthocyanin accumulation. During biotrophic growth, Rsp3 decorates the hyphal surface and interacts with at least two secreted maize DUF26-domain family proteins (designated AFP1 and AFP2). AFP1 binds mannose and displays antifungal activity against the *rsp3* mutant but not against a strain constitutively expressing *rsp3*. Maize plants silenced for AFP1 and AFP2 partially rescue the virulence defect of *rsp3* mutants, suggesting that blocking the antifungal activity of AFP1 and AFP2 by the Rsp3 effector is an important virulence function. Rsp3 orthologs are present in all sequenced smut fungi, and the ortholog from *Sporisorium reilianum* can complement the *rsp3* mutant of *U. maydis*, suggesting a novel widespread fungal protection mechanism.

[1] Department of Organismic Interactions, Max Planck Institute for Terrestrial Microbiology, 35043 Marburg, Germany. [2] Mass Spectroscopy Facility, Max Planck Institute for Terrestrial Microbiology, 35043 Marburg, Germany. [3] Center for Microscopy and Imaging (CMI), Baylor University, Waco, Texas 76798-7046, USA. [4] LOEWE Center for Synthetic Microbiology and Faculty of Chemistry, Philipps-Universität Marburg, 35032 Marburg, Germany. [5] Present address: Department of Pharmacology, Max Planck Institute for Heart and Lung Research, 61231 Bad Nauheim, Germany. [6] Present address: Department of Plant Cell Biology, Albrecht-von-Haller-Institute, Georg-August-University-Göttingen, 37077 Göttingen, Germany. [7] Present address: Bio-Protection Research Centre, Lincoln University, PO Box 64, Lincoln 7647, New Zealand. [8] Present address: Düsseldorfer Straße 177, 45481 Mülheim an der Ruhr, Germany. Correspondence and requests for materials should be addressed to R.K. (email: kahmann@mpi-marburg.mpg.de)

The dimorphic fungus *Ustilago maydis* causes smut disease in maize. Prominent symptoms of the disease are anthocyanin induction and large tumors in which the fungus proliferates and produces spores. The disease cycle is initiated by the dikaryotic hyphae. Upon the perception of surface cues, these hyphae form infection structures that penetrate epidermal cells. During penetration, the host plasma membrane invaginates and surrounds the invading hypha, thus establishing an extended interaction zone characteristic for a biotrophic interaction[1]. *U. maydis* secretes 467 putative effector proteins that contribute to the establishment of the biotrophic stage and tumor formation[1,2]. Such effector proteins function either in the interaction zone (apoplastic effectors) or translocate to plant cells and modify processes inside plant cells (translocated effectors). Many of these effectors lack functional domains and only a small subset is conserved in related smut fungi[1,3]. Up to date, only few of these effectors have been functionally characterized[1]. The two apoplastic effectors Pep1 and Pit2 inhibit peroxidase and cysteine proteases, respectively[4,5]. The three translocated effectors Tin2, Cmu1 and See1 induce anthocyanin formation[6], alter host chorismate metabolism[7], and stimulate cell division associated with tumorigenesis[8], respectively.

Repeat-containing effectors have been identified in several filamentous microbes. They can either associate with microbial surfaces, reside in the interface between microbe and host, or translocate into specific compartments of the host cells[9].

*U. maydis* encodes fifteen effector proteins containing internal repeats[10]. In eight of these proteins, the repeats are shown or predicted to be separated by Kex2-like cleavage sites and thus likely processed while the other seven repeat-containing proteins are not predicted to be processed by Kex2[10,11]. Of these, only the membrane-bound signaling mucin-like protein Msb2 was characterized and shown to regulate appressorium development[12]. In other fungi, many of the non-processed repetitive proteins are attached to the cell wall by glycosylphosphatidylinositol-anchors[13,14]. Their repeat domains are often highly variable in length[15,16], which is considered to result from homologous recombination or slippage during replication[17]. Variations in repeat numbers are thought to provide functional diversity and allow adaptation to environmental changes or escape from host immunity. *Saccharomyces cerevisiae* encodes 44 proteins containing intragenic tandem repeats, and these also show length polymorphisms[15]. In the Flo1p protein, the repeat number is positively correlated with an increase in cell–cell adhesion properties[15]. *Mycosphaerella graminicola* harbors 23 genes predicted to encode surface-associated proteins possessing internal tandem repeats, and isolates vary in intragenic repeat numbers[16]. The genome of *Aspergillus fumigatus* contains 292 genes with internal repeats, and size variation in repeat-containing regions was detected in clinical isolates[18].

In addition, there are repeat-containing effectors that translocate into specific compartments of host cells where they elicit their virulence function. For example, *Colletotrichum graminicola* CgEP1 was detected in the plant nucleus and could bind DNA nonspecifically[19]. *Rhizophagus irregularis* SP7 is presumed to enter the host nucleus and target the transcription factor MtERF19[20]. *Phytophthora infestans* PexRD54 binds the host ATG8CL protein and triggers autophagosome formation[21]. Whether the tandem repeats contribute to these functions is still unknown.

In this study, we functionally analyze the repetitive effector protein Rsp3 (UMAG_03274) of *U. maydis* and provide evidence that this effector has a conserved virulence-promoting function conferred by its ability to shield fungal hyphae from the action of maize antifungal proteins.

## Results

**rsp3 shows length polymorphisms**. Rsp3 belongs to the *U. maydis* effectors without predicted functional or structural domains, and represents a core effector that is present in five sequenced smut fungi[3]. Rsp3 is predicted to be a non-processed repeat-containing protein lacking a GPI anchor. Downstream of the signal peptide, a glutamine and proline-rich region (QP) is followed by a cysteine-rich region (Cys). The long C-terminal domain comprising more than half of the molecule consists of a complex array of several different repetitive units (Fig. 1a).

As the *rsp3* genes in the compatible haploid strains FB1 and FB2[22] showed significant length differences of 2.61 and 2.37 kbp, respectively (Fig. 1a; Supplementary Fig. 1a, b), we analyzed length polymorphisms also in field isolates from different locations in Mexico. The *rsp3* alleles amplified from these strains varied in length between 1.8 and 2.6 kbp. Sequencing revealed that these polymorphisms were caused by deletion or expansion of repeats in the C-terminal domain (Fig. 1a; Supplementary Fig. 1).

A qRT-PCR analysis showed that *rsp3* was not expressed in axenic culture but was specifically and highly induced during the biotrophic interaction (Fig. 1b). Induction was already observed when appressoria develop between 0.5 and 1 day post infection (dpi). *rsp3* expression peaked at 2 dpi and then gradually decreased (Fig. 1b). A recent time-resolve transcriptome analysis placed *rsp3* into an effector-enriched module associated with the establishment of biotrophy[23], i.e., the early stages of fungal development inside the plant.

**Rsp3 is an important virulence factor in *U. maydis***. To investigate a contribution to virulence, *rsp3* was deleted in the solo-pathogenic haploid strain SG200[2] and its derivative SG200AN1 expressing a *UMAG_01779* promoter-GFP fusion protein in cells developing appressoria[24,25]. To study early infection-related development, appressoria formation and penetration were analyzed microscopically and quantified 18 h post infection. Appressorium formation as well as penetration of SG200AN1Δrsp3 occurred with comparable efficiency to SG200AN1 (Supplementary Fig. 2a, b).

SG200Δrsp3 strains showed a strong attenuation of virulence and both tumor numbers as well as tumor size were significantly reduced compared to SG200 (Fig. 2a, b). In addition, the mutant failed to induce anthocyanin (Fig. 2b). These phenotypes could be rescued by complementation with a single copy of the *rsp3* allele of FB1 or the shortest *rsp3* allele from Toluca-6 (T6) (Fig. 2a, b). *rsp3* alleles carrying HA-tags at either the N terminus downstream of the signal peptide or at the C terminus could fully complement the virulence phenotype of the Δ*rsp3* mutant (Fig. 2a, b). However, a *rsp3* allele carrying substitution of all nine cysteine residues to alanine ($rsp3_{9CA}$) in the Cys-rich domain (Fig. 1a) or truncated alleles lacking the Cys-rich domain ($rsp3_{\Delta Cys}$; Δ244–333 amino acids) could not complement the virulence phenotype of the Δ*rsp3* mutant (Fig. 2a, b) while anthocyanin production was observed (Fig. 2b). Orthologs of Rsp3 are found in all sequenced smut species and display a relatively high conservation in their N-terminal domains (35–55% identity) while the repetitive domains show only 4–21% identity (Supplementary Fig. 3). The *S. reilianum* ortholog SrRsp3, which has only 34% sequence identity to *U. maydis* Rsp3 (FB2), could fully complement the virulence phenotype of the *U. maydis rsp3* mutant (Fig. 2c).

Next, we investigated plant colonization by SG200Δrsp3 by confocal microscopy and compared this to colonization by SG200. Reduced proliferation of Δ*rsp3* strains was observed at all stages of the infection and mutant hyphae remained intracellularly and did not proliferate along the veins where wild type hyphae accumulated and were found to spread from 4 dpi

onwards (Fig. 2d; Supplementary Fig. 2c). Compared to the *rsp3* mutant and SG200, SG200AN1Δrsp3-rsp3$_{ΔCys}$ showed an intermediate phenotype characterized by a higher degree of colonization and some hyphal growth along the veins (Supplementary Fig. 2d). These results illustrate that Rsp3 is an important virulence factor.

**The N terminus of Rsp3 is processed.** The first 23 amino acids of Rsp3 are predicted by SignalP 4.1[26] to be a signal peptide. To

visualize secretion, we generated SG200Δrsp3 strains, which constitutively expressed the biologically active Rsp3-HA or a protein with an additional Myc-tag inserted between amino acid 46 and 47, Myc$_{47}$-Rsp3-HA (Fig. 3a). In supernatants of these strains, Rsp3-HA and Myc$_{47}$-Rsp3-HA could be detected by western blot developed with an anti-HA antibody (Fig. 3b). Surprisingly, we failed to detect Myc$_{47}$-Rsp3-HA in the supernatant fraction with an anti c-Myc antibody (Fig. 3b), despite being able to detect the protein with this antibody in the cell pellet fraction (Fig. 3b). This indicates that Rsp3 is processed at the N

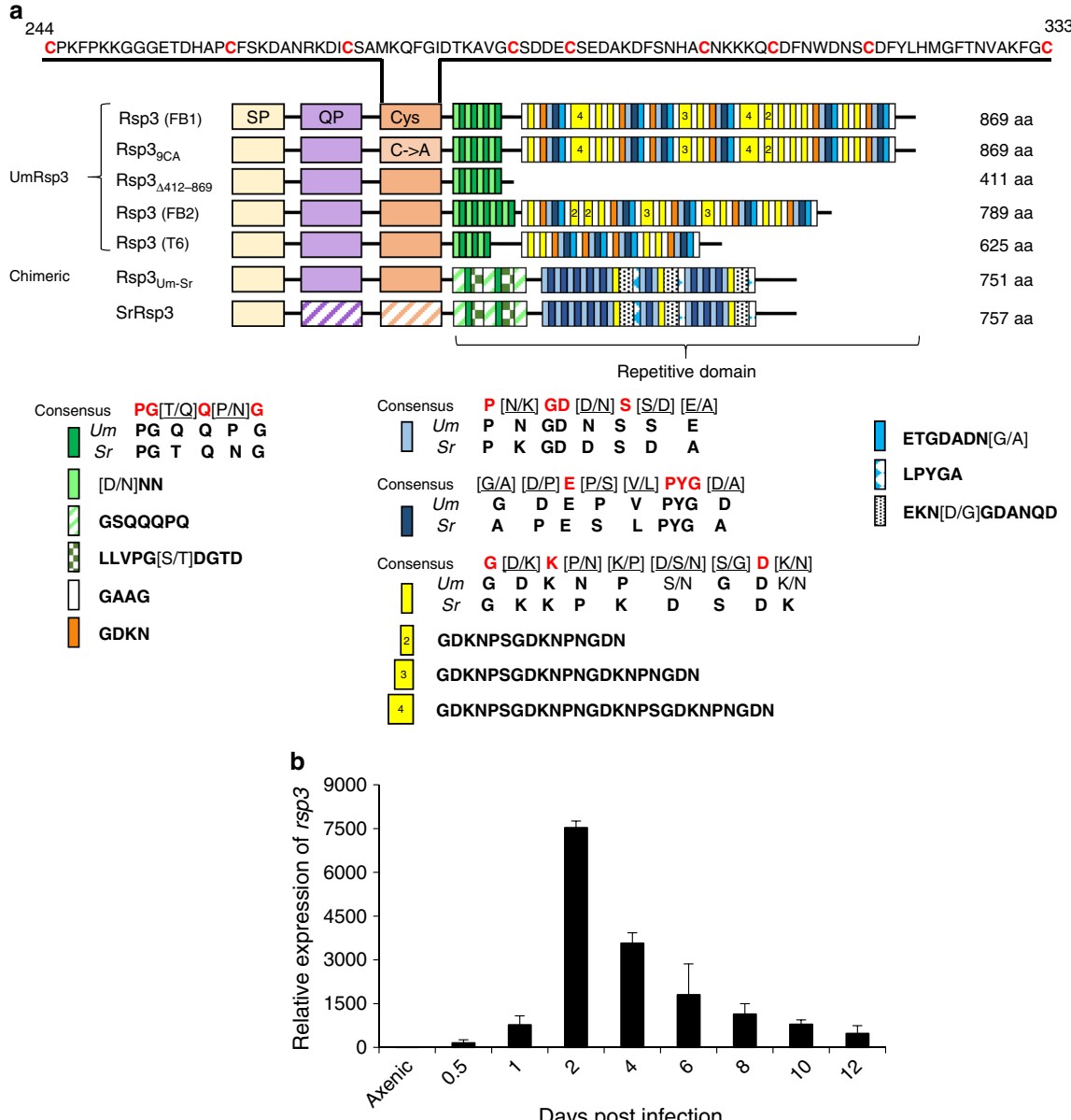

**Fig. 1** Rsp3 shows length polymorphisms in field isolates and is induced after plant colonization. **a** The domain architecture of Rsp3 from *U. maydis* (Um) strains (FB1, FB2, and Toluca-6) and *S. reilianum* (Sr) as well as of C terminally truncated (Δ412–869), the 9CA mutant protein and chimeric proteins is depicted. aa amino acids, SP signal peptide, Cys cysteine-rich region (amino acids 244–333). This region is shown enlarged above. QP: glutamine and proline rich region. The different types of repeats are colored using the code shown below. The four degenerate repeats detected in *U. maydis* and *S. reilianum* have the same colors. Invariant amino acids in the consensus sequence are highlighted in red. Numbers 2, 3, and 4 in the yellow repeat indicate overlapping extensions of the short yellow repeat. **b** Expression analysis of *rsp3* by quantitative RT-PCR. Maize plants were infected with a mixture of FB1xFB2 and segments of infected leaves were harvested at the indicated time points. RNA was prepared and subjected to qRT-PCR. RNAs prepared from FB1 and FB2 grown in YEPSL were mixed to provide the sample labeled axenic culture. Expression levels of *rsp3* were normalized relative to the constitutively expressed peptidyl-prolyl isomerase (*ppi*). The expression level of *rsp3* in axenic culture was set to 1.0. Three biological replicates were analyzed. Values represent mean ± sd

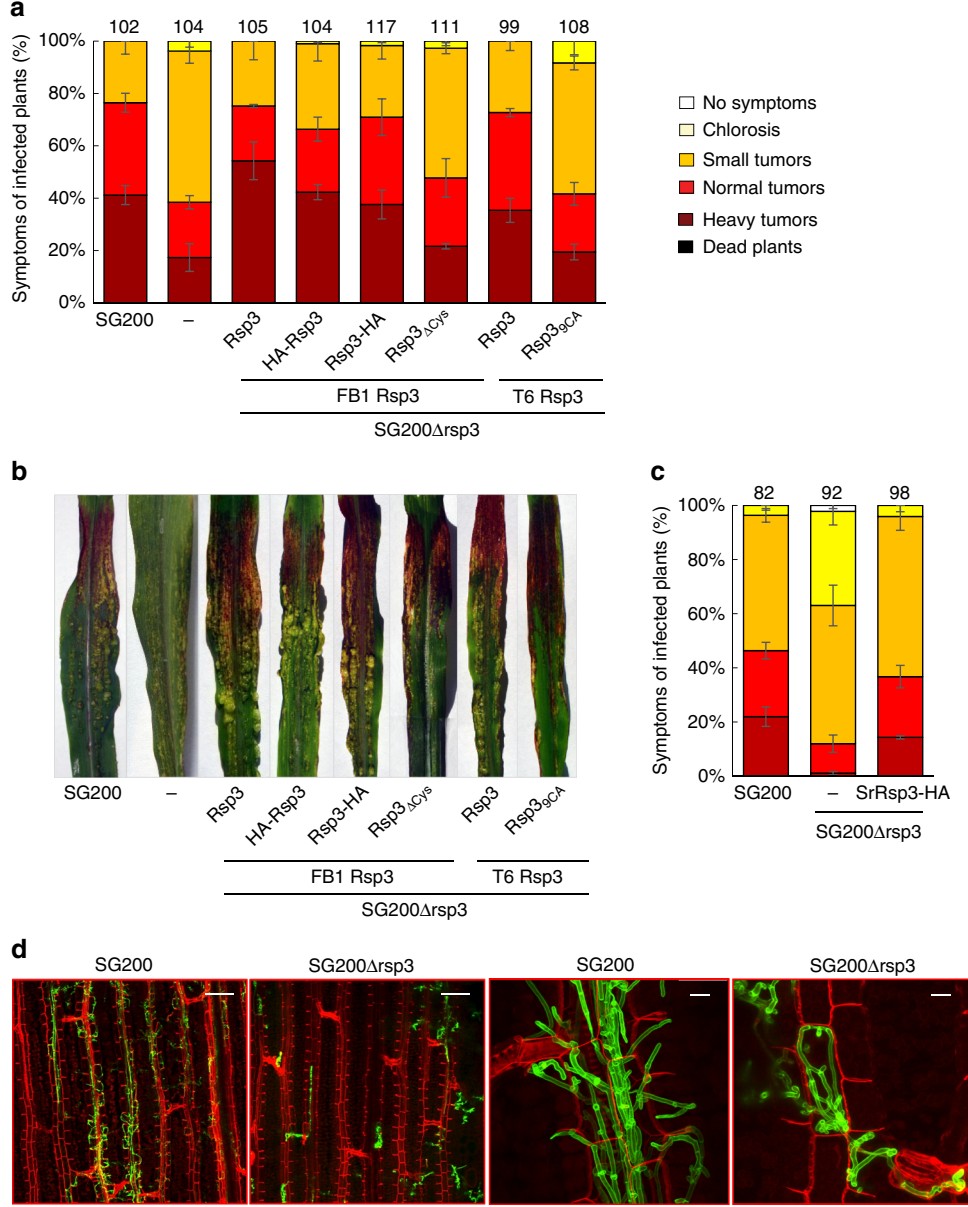

**Fig. 2** Rsp3 is required for virulence of *U. maydis*. **a** 7-day-old-maize seedlings were infected with the indicated strains and disease symptoms were scored at 12 dpi. Except for SG200 all other strains are derived from SG200Δrsp3 and express either the FB1 or Toluca-6 Rsp3 or SrRsp3-HA proteins under control of the FB1 derived *rsp3* promoter. Total numbers of infected plants from the three biological replicates are indicated above the respective columns. Error bars indicate standard deviations for each symptom category. **b** Representative pictures of disease symptoms on maize leaves infected by the indicated strains after 12 dpi. **c** Maize seedlings were infected with the indicated strains and scored as described in **a**, using the symptom categories depicted in **a**. **d** Maize seedlings infected by the indicated *U. maydis* strains were observed at 4 dpi by confocal microscopy. Fungal hyphae were stained with WGA-AF488 (green). Plant cell walls were stained with propidium iodide (red). Bars: 100 μm (two leftmost panels); 25 μm (two rightmost panels)

terminus. We also noticed that Rsp3-HA did not migrate according to its expected size (88 kD) but had an apparent molecular weight of about 150 kD (Fig. 3b). As Rsp3-HA did not show *N*- or *O*-glycosylation (Supplementary Fig. 2e), we assume that the anomalous migration of Rsp3 is due to intrinsic properties of the protein. Using the D2P2 database (http://d2p2.pro/search) and the ProtParam tool (http://web.expasy.org/protparam/), Rsp3 is predicted to be negatively charged and intrinsically unstructured with two disordered regions located between amino acids 23–265 and amino acids 330–869. Negatively charged amino acids and disordered structures have been reported to contribute to anomalous migration behavior[27–29],

making it likely that these features explain the anomalous migration of Rsp3.

To determine the processing site in Rsp3, Rsp3-HA purified from the supernatant of SG200Δrsp3-P$_{otef}$-rsp3-HA was subjected to N-terminal sequencing by Edman degradation. This allowed to identify DGGA as the four N-terminal amino acids of the secreted protein suggesting that processing occurs between amino acids 60 and 61 (Fig. 3a). To determine whether the region between signal peptide and N terminus of the processed Rsp3 is relevant for function, we produced a truncated version of Rsp3-HA lacking the region between amino acids 24 and 60 (Rsp3$_{Δ24-60}$-HA; Fig. 3a) and examined its secretion. Surprisingly, this

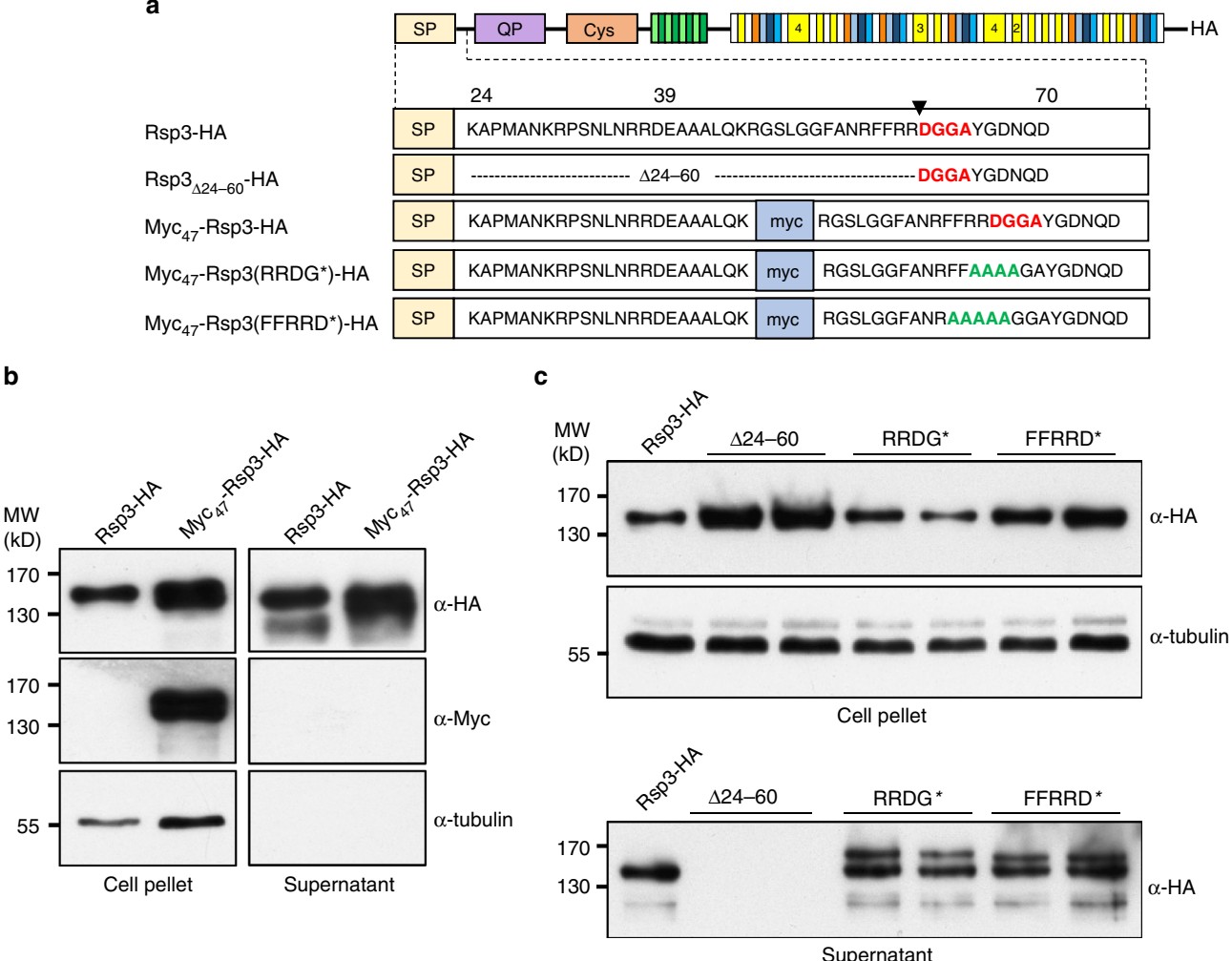

**Fig. 3** The N terminus of Rsp3 contains signals for processing and secretion. **a** Schematic representation of constructs used to analyze N-terminal processing of Rsp3. In all constructs, expression was driven from the constitutive *otef* promoter and all contain a C-terminal HA-tag. Amino acid sequences from 24 to 70 downstream of the signal peptide (SP) are shown. The position of Myc-tag insertion is depicted in light blue. Amino acids DGGA identified by N-terminal sequencing as N terminus of secreted Rsp3 are indicated in red. The arrow indicates the cleavage site. Amino acids subjected to alanine substitution are indicated in green. **b** Western blot analysis of Rsp3 secretion. SG200Δrsp3 strains expressing the indicated proteins were grown in CM liquid medium to an $OD_{600}$ of 0.6. Proteins from cell pellets and from supernatants (collected after TCA precipitation) were prepared subjected to western blot. The western blots were developed with either anti-HA or anti c-Myc antibodies as indicated. Detection of tubulin via an anti-tubulin antibody served as internal control for a cytosolic protein. **c** Secretion of Rsp3 variants carrying amino acid substitutions or deletions in the N-terminal domain. SG200Δrsp3 strains expressing the indicated proteins were analyzed by western blot after fractionation in supernatant and cell pellet as in **b**. Proteins shown in western blot analysis are derived from two independent clones. Experiments were repeated two times for Fig. 3b and three times for Fig. 3c, and one representative experiment is shown. Full blots are shown in Supplementary Fig. 11

deletion abolished secretion and Rsp3Δ24-60-HA accumulated inside the cells (Fig. 3c, upper panel). It could be that cleavage of the signal peptide does not occur when the signal peptide is fused differently to the disordered N-terminal domain, although other possibilities are not excluded. Next, we attempted to abolish processing by generating alanine substitution of amino acids in the vicinity of the cleavage site (Fig. 3a; Myc47-Rsp3(RRDG*)-HA and Myc47-Rsp3(FFRRD*)-HA). Both mutant proteins were still secreted (Fig. 3c, lower panel) and could complement the virulence phenotype of the Δrsp3 strain (Supplementary Fig. 4a, b). However, compared to Rsp3-HA protein, the molecular weights of both mutant proteins was higher and two discrete protein species were detected in the supernatant fraction (Fig. 3c, lower panel). This indicates that the introduced substitutions have eliminated the identified cleavage site between amino acids

60 and 61, but retained additional cleavage sites further upstream. These results suggest that N-terminal processing of Rsp3 is linked to its secretion and most likely to its function.

**Rsp3 decorates the surface of fungal hyphae**. To assess how Rsp3 contributes to virulence of *U. maydis*, the effector was localized. Plant samples infected with either SG200 or SG200Δrsp3-rsp3-HA, which produces biologically active Rsp3-HA under its native promoter, were subjected to immunogold labeling and the distribution of gold particles was quantified (Supplementary Table 1). While labeling of SG200-infected tissue revealed low and non-specific background in most compartments (Fig. 4a, top panel), Rsp3-HA was primarily detected inside the fungal cytosol and in the biotrophic interface but rarely inside plant cells (Fig. 4a, lower panel; Supplementary Table 1).

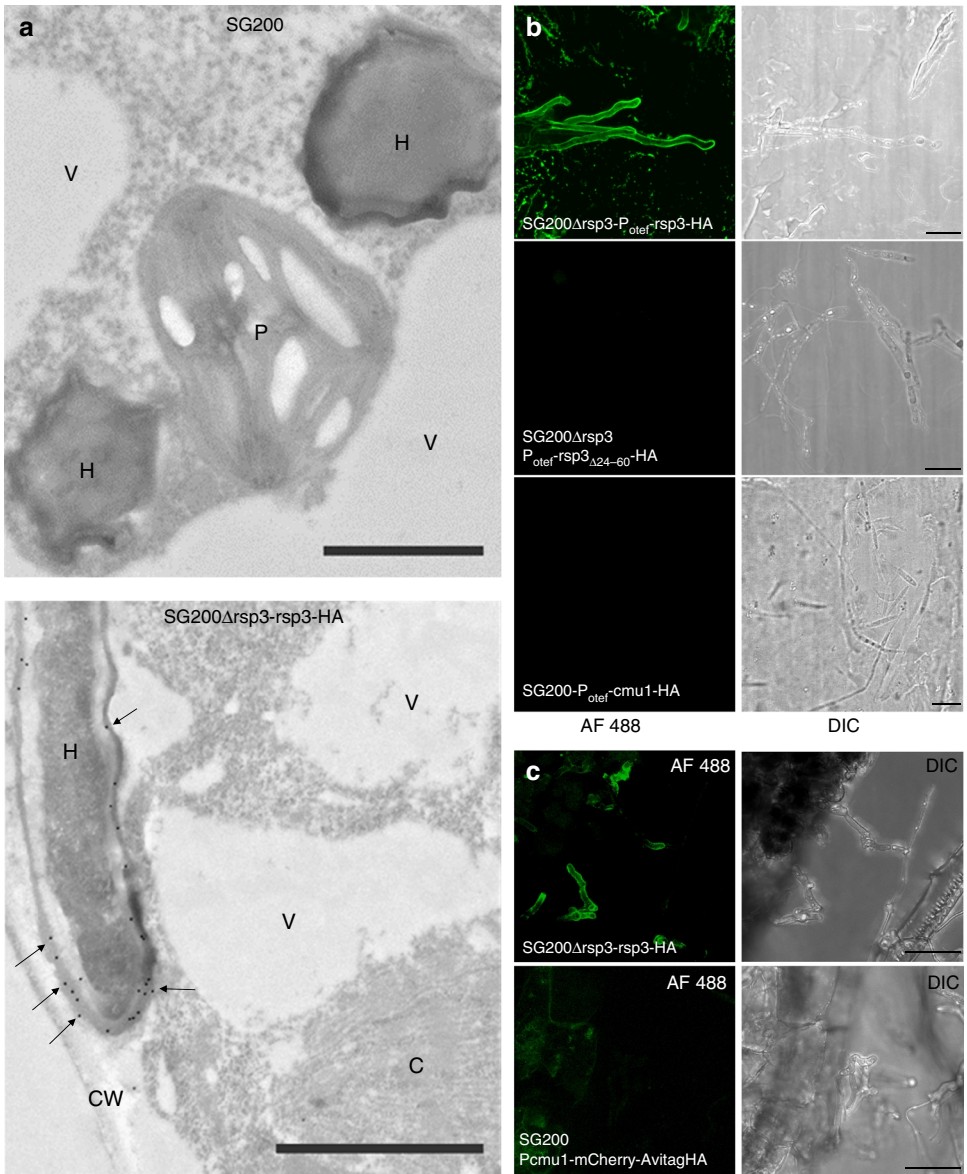

**Fig. 4** Secreted Rsp3-HA binds to the *U. maydis* cell wall. **a** Immunogold labeling of Rsp3-HA in the biotrophic interface of infected maize leaves harvested at 4 dpi. Transmission electron micrographs showing immunogold labeling (arrows) of *Zea mays* infected with *U. maydis* strains SG200 and SG200Δrsp3-rsp3-HA. H hyhae, CW cell wall, P plastid, V vacuole, C chloroplast. Bars = 1 μm. Gold particles from several images of embedded hyphae from a single experiment were quantified separately for each cellular compartment ($n \geq 20$ for each cellular compartment; Supplementary Table 1). **b** Secreted Rsp3-HA binds to the fungal cell wall in filaments grown on artificial surface. All indicated strains were treated with hydroxy-fatty acids and sprayed on Parafilm M to induce filamentation. Cells were immunostained with an anti-HA antibody and an AF488-conjugated secondary antibody without prior permeabilization. SG200Δrsp3-P$_{otef}$-rsp3(Δ24–60)-HA serves as a negative control since Rsp3-HA(Δ24–60) is not secreted. SG200-P$_{otef}$-cmu1-HA serves as control for a secreted effector that does not bind to fungal cell wall. In the left panel, Alexa Fluor 488 (AF488) fluorescence is shown while the right panel shows the respective DIC images. Bars = 10 μm. **c** Rsp3-HA binds to the fungal hyphae during plant colonization. Maize leaves infected with either SG200Δrsp3-rsp3-HA or SG200-P$_{cmu1}$-mCherry-AvitagHA expressing cytosolic mCherry-AvitagHA were collected at 3 dpi, partially macerated, fixed, and immunostained with an anti-HA antibody and an AF488-conjugated secondary antibody without prior permeabilization of the fungal cells. Bars: 25 μm. The experiments **b** and **c** were repeated two times and one representative repeat is shown

As an accumulation in the biotrophic interface could be a sign of attachment to fungal hyphae, we next investigated this possibility using immunofluorescence microscopy. SG200Δrsp3-derived strains constitutively expressing either Rsp3-HA, non-secreted Rsp3$_{Δ24-60}$-HA or Cmu1-HA, an effector that is translocated to plant cells[7], were stimulated with hydroxy-fatty acids and sprayed on Parafilm M to induce filamentation. Non-permeabilized cells were subjected to anti-HA immunostaining using an anti-HA antibody followed by a secondary antibody conjugated to Alexa Fluor 488. We detected fluorescence signals around hyphae of SG200Δrsp3-P$_{otef}$-rsp3-HA but not around hyphae, which failed to secrete Rsp3$_{Δ24-60}$-HA or hyphae secreting Cmu1-HA (Fig. 4b). In areas on Parafilm M, where the SG200Δrsp3-P$_{otef}$-rsp3-HA strain had not differentiated filaments but grew by budding, strong fluorescence was detected on the surface (Supplementary Fig. 4c). This demonstrates that constitutively expressed Rsp3-HA can attach to the surface of budding as well as filamentous cells of *U. maydis*.

To investigate whether Rsp3 when expressed from its native promoter attaches to fungal hyphae during colonization, leaf

samples infected by SG200Δrsp3-rsp3-HA or SG200 P$_{cmu1}$-mCherry-AvitagHA[30] expressing cytosolic mCherry-AvitagHA from the *cmu1* promoter were subjected to anti-HA immuno-staining after partially macerating the infected plant tissue. Rsp3-HA was detected around the outside of fungal hyphae (Fig. 4c, upper panel) while non-secreted mCherry-AvitagHA could not be detected (Fig. 4c, lower panel).

To determine whether Rsp3 might also reside as non-attached form in the biotrophic interface, we analyzed apoplastic fluid from maize leaves infected with SG200Δrsp3-rsp3-HA and SG200-cmu1-AvitagHA at 3 dpi. Although Cmu1-AvitagHA was detected by western blot in TCA-precipitated apoplastic fluid samples, we failed to detect soluble Rsp3-HA (Supplementary Fig. 4d).

As Rsp3-HA can bind to the surface of budding cells and filamentous hyphae of *U. maydis* when it is constitutively expressed, it is conceivable that Rsp3 might bind to cell wall components also when added externally. To test this, we purified Rsp3-HA from the supernatant fraction of SG200Δrsp3-P$_{otef}$-rsp3-HA. When incubated with budding cells of SG200 grown in liquid medium and hyphae of SG200 generated on parafilm after spraying, i.e., cells which do not express *rsp3* at this stage, fluorescence signals were mostly restricted to tips and septa, and were sometimes surrounding filaments (Supplementary Fig. 5). To test whether Rsp3 binds to the specific cell wall components of *U. maydis*, we tested binding to *Colletotrichum graminicola* (CgM2) conidia and budding cells of *S. cerevisiae* AH109. Binding of added Rsp3-HA could not be detected (Supplementary Fig. 6a, b). This suggests that Rsp3 might attach to cell wall components, which are *U. maydis* specific, and efficient cell wall association may require that the protein is synthesized by the fungus. Alternatively, cell wall components required for the binding of Rsp3 may not be present or accessible in conidia of *C. graminicola* or budding cells of *S. cerevisiae*. In fungal pathogens, infection-related development and plant colonization have been shown to involve coordinated changes in cell wall composition to escape recognition by the immune system[31] and such alterations might be needed to allow binding of Rsp3.

**Rsp3 interacts with secreted maize DUF26-domain proteins**. Having shown that Rsp3 is attached to biotrophic hyphae of *U. maydis*, we expected putative interaction partners from maize to be secreted apoplastic proteins. To identify interaction partners of Rsp3, we purified Rsp3-HA from culture supernatants of strain SG200Δrsp3-P$_{otef}$-rsp3-HA. The purified protein was mixed with apoplastic fluid from SG200 or mock inoculated plants. After affinity purification on HA agarose beads, bound proteins were eluted and analyzed by SDS-PAGE. A band with an approximate molecular weight of 25 kD was more intense in the sample of Rsp3-HA incubated with apoplastic fluid of SG200-infected plants than in the sample of mock inoculated plants (Supplementary Fig. 7a). By mass spectrometry analysis from two independent experiments, the bands were shown to contain maize proteins, GRMZM2G043878 and GRMZM2G334181, while these were absent in the mock samples. These two proteins which we designate AFP1 and AFP2 (Anti-Fungal Protein 1 and 2) share 42% amino acid identity and are both predicted to be secreted (Fig. 5a).

To confirm the direct interaction of Rsp3 with AFP1 protein, AFP1-His transiently expressed in *Nicotiana benthamiana* was purified using Ni-NTA-agarose beads. Purified AFP1-His was incubated with culture supernatants containing Rsp3-HA, Rsp3$_{9CA}$-HA, Rsp3$_{Δ412-869}$-HA lacking the C-terminal 458 amino acids, or a chimeric Rsp3$_{Um-Sr}$-HA protein in which the N-terminal domain of UmRsp3 is fused with the C-terminal repetitive domain of SrRsp3 (Fig. 1a). AFP1-His could pull down

Rsp3-HA, Rsp3$_{9CA}$-HA, and chimeric Rsp3$_{Um-Sr}$-HA (Fig. 5b, c). However, Rsp3$_{Δ412-869}$-HA poorly interacted with AFP1-His (Fig. 5b), suggesting that the interaction with AFP1 is largely occurring through the C-terminal repetitive domain of Rsp3 and likely does not involve the cystein-rich domain. AFP2-His purified from *N. benthamiana* also interacted with Rsp3-HA (Supplementary Fig. 7b). Rsp3$_{9CA}$-HA and Rsp3$_{Δ412-869}$-HA were still able to attach to the surface of fungal hyphae when constitutively expressed (Supplementary Fig. 4c), suggesting that binding is mediated by the N-terminal domain of Rsp3. Although Rsp3$_{Um-Sr}$-HA showed full biological activity, Rsp3$_{Δ412-869}$-HA was only partially able to complement the virulence phenotype of an *rsp3* deletion strain (Supplementary Fig. 4b).

**Maize AFP1 protein binds mannose**. BLASTP searches revealed that maize AFP1 and AFP2 proteins have sequence similarity to Gnk2, a secreted antifungal mannose binding protein from *Ginkgo biloba* seeds[32] containing a DUF26 domain (C-X$_8$-C-X$_2$-C motif) (stress-antifungal domain PF01657; http://pfam.sanger.ac.uk/family/PF01657) (Fig. 5a). Maize B73 contains 45 DUF26 family members, which form 14 classes based on their predicted domain architecture (Supplementary Fig. 8a). Ten of the DUF26 family genes are predicted to encode secreted proteins. AFP1 and AFP2 proteins belong to class 5 whose members contain a signal peptide and two DUF26 domains but lack other domains (Supplementary Fig. 8a). By qRT-PCR analysis, AFP1 was shown to be highly induced during infection by *U. maydis* with induction being highest after 12 h and then returning to a lower level (Supplementary Fig. 9a). On the basis of a time-resolved RNAseq data set of *U. maydis* infected plants[23], AFP1 was the most highly induced DUF26 gene of maize followed by AFP2 (Supplementary Fig. 9b, c). Of the eight DUF26 genes predicted to encode secreted proteins (beside AFP1 and APF2), GRMZM2G365282 was not expressed under our infection conditions and the remaining seven genes were either not induced or induced weakly (Supplementary Fig. 9d).

On the basis of amino acid sequence alignments (Fig. 5a) and modeling based on the structure of Gnk2[32] (Supplementary Fig. 8b), AFP1 appears to be composed of two copies of Gnk2 and each of the Gnk2-homology domains of AFP1 consists of two α-helices and five β-sheets. The cysteine residues in the DUF26 domain are modeled to form three disulfide bonds bridging secondary structure elements critical for stabilization of AFP1. On the basis of the modeling of AFP1, residues S34, R115, and E126 in the N terminus and N144, Q227, and E238 in the C terminus are corresponding to residues N37, R119, and E130 in Gnk2. In Gnk2, these three residues are involved in mannose binding and confer the antifungal activity[32] (Fig. 5a; Supplementary Fig. 8b, c).

To investigate whether AFP1 can bind mannose, we expressed AFP1-His proteins with or without alanine substitutions in the putative mannose binding sites in *N. benthamiana*. The purified AFP1-His proteins were then tested for their ability to bind to mannose-agarose beads. While strong binding was observed for wild type AFP1-His, binding was reduced for AFP1*-His (alanine substitutions in the putative mannose-interacting amino acids in the N terminus) and was undetectable for AFP1**-His (alanine substitutions in all six putative mannose-interacting amino acids) (Fig. 6a). This indicates that maize AFP1 is a mannose binding protein with two mannose binding sites. An alignment of all secreted DUF26-domain proteins of maize revealed that amino acids putatively involved in mannose-binding are detected in several other family members beside AFP1 (Supplementary Fig. 8c).

**Rsp3 blocks antifungal activity of maize AFP1**. As Rsp3 is bound to the surface of fungal hyphae and interacts with maize

AFP proteins, we speculated that Rsp3 might act as a shield for fungal hyphae and protect them against a potential antifungal activity of AFP1. To test whether AFP1 has antifungal activity, we monitored the survival of SG200Δrsp3 after incubation with AFP1-His and AFP1**-His. A cell suspension ($OD_{600} = 0.001$) was incubated with AFP1 proteins for 3 h and plated on PD-

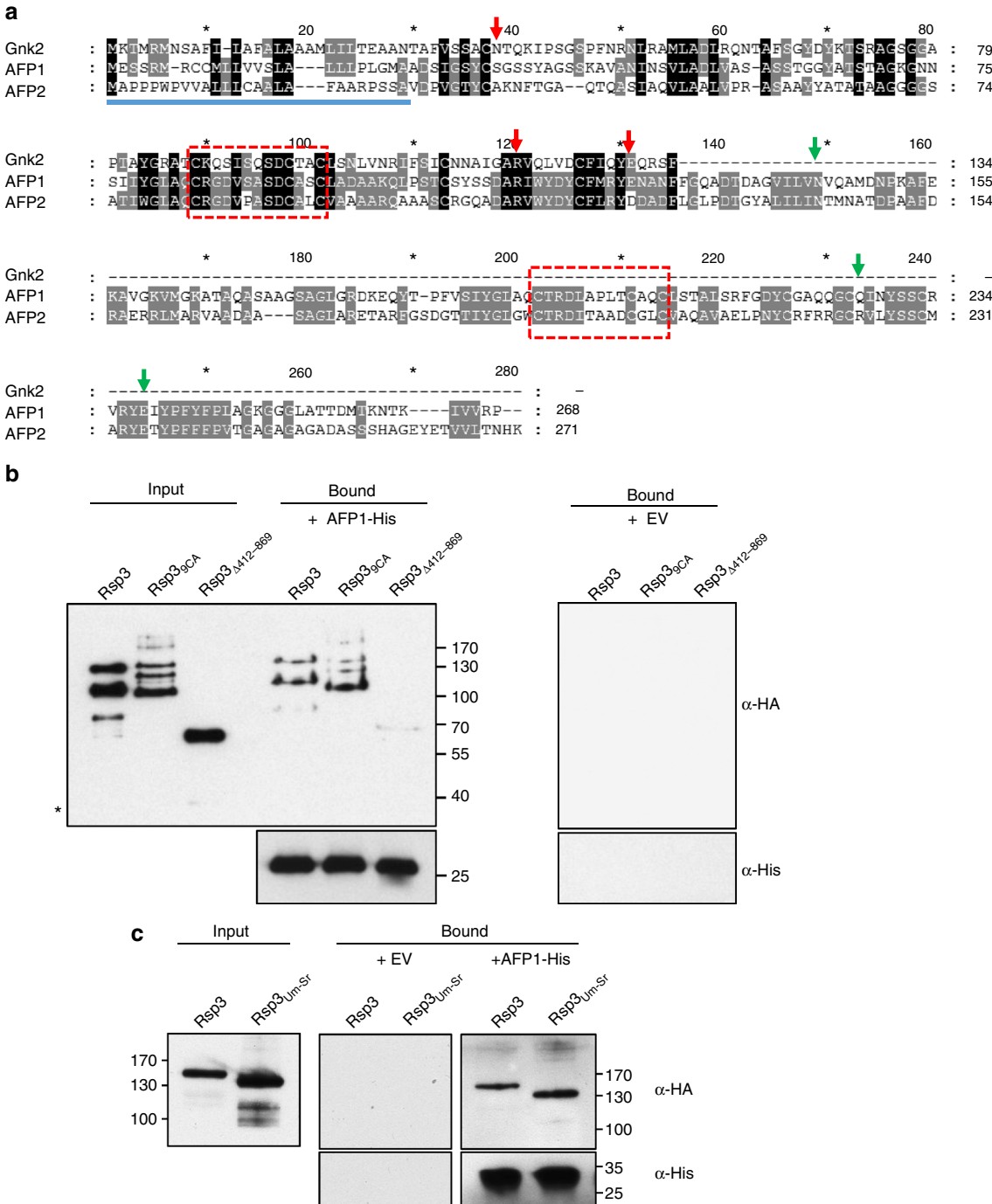

**Fig. 5** Rsp3 interacts with maize secreted AFP1 protein. **a** Amino acid sequence alignment of maize AFP1, APF2, and *G. biloba* Gnk2. The signal peptide is underlined with a blue line. Conserved amino acids are highlighted in black. The DUF26 domains ($C-X_8-C-X_2-C$) are indicated by red dashed boxes. The red and green arrows indicate residues putatively involved in mannose binding in the N- or C-terminal domains of AFP1 and APF2, respectively. **b** Secreted Rsp3 interacts with purified AFP1. Leaf tissues of *N. benthamiana* infiltrated with *A. tumefaciens* carrying either a AFP1-His expressing plasmid or an empty vector (EV) as negative control were subjected to NTA-affinity purification. Prior to protein elution the NTA-agarose beads were mixed with culture supernatants of SG200Δrsp3 expressing the indicated Rsp3-HA variants under control of the *otef* promoter. The input and bound proteins were detected by western blot using anti-His-HRP and anti-HA antibodies. The asterisk (*) indicates a truncated form of the Rsp3$_{Δ412-869}$-HA. **c** Secreted Rsp3$_{Um-Sr}$ hybrid protein interacts with purified AFP1. Culture supernatants of SG200Δrsp3 expressing the indicated Rsp3-HA variants were incubated with AFP1-His as described in **b** and interaction was shown by western blot as in **b**. The experiments in **b** and **c** were repeated three times and one representative experiment is shown. Full blots are shown in Supplementary Fig. 12

plates. SG200Δrsp3 showed lower survival after treatment with AFP1-His than with AFP1**-His (Fig. 6b). A graphical presentation of results from three biological repeats revealed a significant difference in titer between the *rsp3* mutant treated with AFP1-His and AFP1**-His (Fig. 6c). This suggests that AFP1 has antifungal activity and mannose binding is required for this activity. To elucidate whether Rsp3 interferes with the presumed antifungal activity of AFP1, we compared the survival of SG200Δrsp3 and SG200Δrsp3-P$_{otef}$-rsp3-HA, a strain constitutively expressing Rsp3-HA and exposing the protein on its surface (Supplementary Fig. 4c). When incubated with AFP1, we observed a significantly higher plating efficiency of the strain constitutively expressing Rsp3-HA compared to the *rsp3* mutant (Fig. 6b, c), suggesting that Rsp3 can protect hyphae against the antifungal activity of maize AFP1. We further tested the presumed antifungal activity of AFP1 by staining the treated cells with the cell death stain SYTOX Orange[33]. In SG200Δrsp3, the percentage of dead cells was about 83% in presence of AFP1-His

versus 19% in the presence of AFP1**-His (Fig. 6d, e), confirming the antifungal activity of AFP1.

**Silencing of maize AFP1 and AFP2 genes enhances virulence.** Having shown that Rsp3 protects hyphae against AFP1, we investigated the biological relevance of this protective effect. To this end, we employed the Foxtail mosaic virus (FoMV) system[34] to downregulate the expression of maize AFP1 and AFP2 genes simultaneously. In two independent experiments, plants silenced for both AFP1 and AFP2 were significantly more susceptible to SG200Δrsp3 infection than plants that had received the empty vector (Fig. 7a). qRT-PCR analysis of individual plants revealed that silencing of AFP1 had occurred in the majority of FoMV-infected plants that had received the silencing constructs (Supplementary Fig. 10a). However, expression of AFP2 was highly upregulated in some of the FoMV-infected plants carrying the silencing constructs (Supplementary Fig. 10b), uncovering some

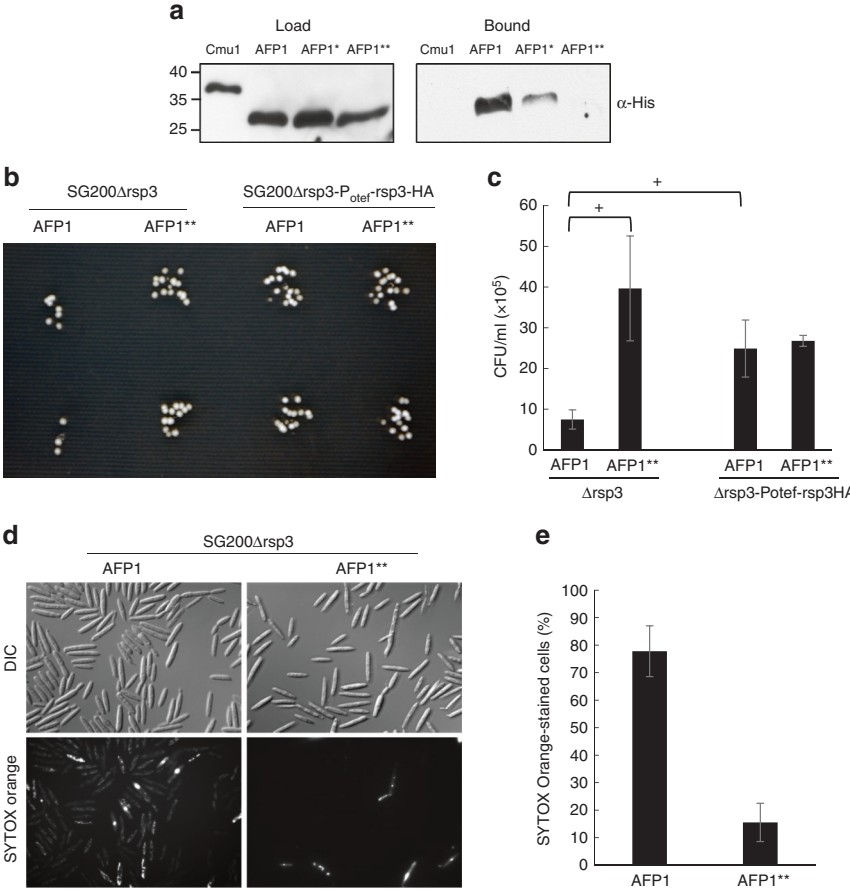

**Fig. 6** Rsp3 blocks the antifungal activity of maize secreted mannose binding protein AFP1. **a** Mannose binding of AFP1 and mutant AFP1 variants. AFP1-His, AFP1*-His (S34A, R115A, and E126A) and AFP1**-His (S34A, R115A, E126A, N144A, Q227A, and E238A) purified from *N. benthamiana* were incubated with mannose-agarose beads. Bound proteins were analyzed by western blot using anti-His-HRP antibodies. Cmu1-His expressed and purified from *E. coli* BL21 served as negative control. The experiment was preformed three times and one representative experiment is shown. Full blots are shown in Supplementary Fig. 13a. **b** AFP1 has antifungal activity. The indicated strains were grown to an OD$_{600}$ of 0.6 and subsequently diluted to OD$_{600}$ = 0.001. Cells were incubated for 3 h with either AFP1-His or AFP1**-His protein. The cell suspensions were spotted twice on a PD agar plate and incubated at 28 °C for 2 days until colonies appeared. The experiment was performed in three biological replicates and one representative experiment is shown. The full photograph is shown in Supplementary Fig. 13b. **c** Quantification of the antifungal activity of AFP1. The survival of the indicated *U. maydis* cells treated with AFP1 or AFP1** was quantified by counting colony forming units (CFUs) from **b**. Data represent mean ± sd of the three biological replicates. *p*-values were calculated by Student's *t*-test, and significant differences (*p* < 0.05) are indicated by +. **d** SYTOX Orange staining reveals cell death inducing ability of AFP1. SG200Δrsp3 was incubated with either AFP1-His or AFP1**-His for 3 h before staining with SYTOX Orange. SYTOX Orange-stained cells were visualized by epifluorescence microscopy. **e** The experiment shown in **d** was done in three biological replicates. About 250 cells were evaluated in each experiment. The percentage of cells stained with SYTOX Orange is indicated. Values represent mean ± sd

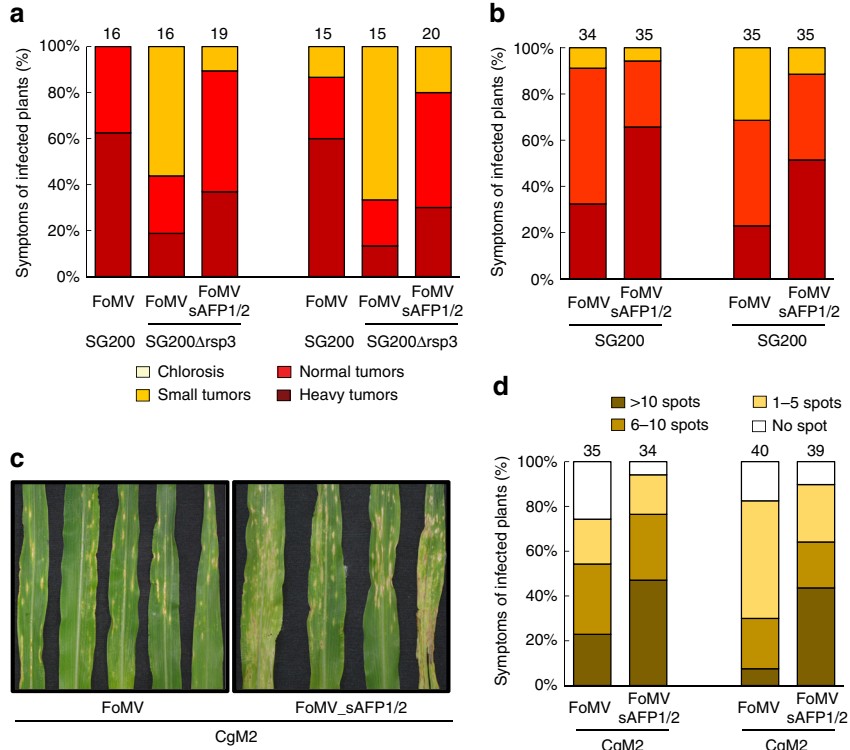

**Fig. 7** Silencing of maize AFP1 and AFP2 genes enhances virulence. Maize seedlings were pre-inoculated with FoMV viral sap expressing AFP1 and AFP2 silencing constructs (FoMVsAFP1/2) or FoMV viral sap without silencing constructs as control (FoMV) for 5 days. **a** Silencing of AFP1 and AFP2 in maize enhances virulence of SG200Δrsp3. FoMVsAFP1/2 or FoMV-treated plants were infected with the indicated strains. Disease symptoms were scored at 12 dpi following the scheme depicted below. **b** Silencing of AFP1 and AFP2 in maize enhances virulence of SG200. FoMVsAFP1/2 or FoMV-treated plants as in **a** were infected with SG200. Disease symptoms were scored at 12 dpi following the scheme depicted in **a**. **c** Macroscopic symptoms of *C. graminicola* on maize plants silenced for AFP1 and AFP2. FoMVsAFP1/2 and FoMV-treated plants were infected with *C. graminicola* CgM2, and symptoms were imaged at 7 dpi (**d**). Quantification of *C. graminicola* leaf spots on maize plants silenced for AFP1 and AFP2. Plants were infected as in **c**. The number of CgM2 spots on the 3rd or 4th leaf were counted and categorized according to the four disease categories given above. Numbers of total infected plants are indicated above the respective columns. Two biological replicates are shown

kind of crosstalk between AFP1 and AFP2 genes. This might explain why rescue of the virulence phenotype of the *rsp3* mutant was only partial. Plants silenced for AFP1 and AFP2 were also significantly more susceptible to *U. maydis* SG200 (Fig. 7b) and *C. graminicola* infection (Fig. 7c, d), than plants that had received the empty vector illustrating that AFP1 and AFP2 target several plant pathogenic fungi.

## Discussion

In this work, we demonstrate that the repetitive secreted effector Rsp3 of *U. maydis* strongly contributes to virulence. Rsp3 counteracts plant defense responses by shielding hyphae from antifungal maize proteins.

Rsp3 protein is not predicted to possess a GPI anchor but is nevertheless associated with the surface of hyphae through its N-terminal domain. A mutant Rsp3 protein with cysteine to alanine substitutions in the cysteine-rich domain can still attach to the hyphal surface (and bind AFP1 weakly) but does not complement the virulence phenotype of the *rsp3* mutant. It is therefore likely that the cysteine-rich region contributes to structural integrity of Rsp3, which is needed for its shielding function.

Although uniform association with hyphae or budding cells was seen when the proteins were expressed by *U. maydis*, added purified Rsp3-HA associated primarily with actively growing regions, i.e., hyphal tips and cell poles as well as septa. This most likely indicates that cell wall components bound by Rsp3 are

accessible during growth or when septation occurs but may be otherwise shielded by other surface-associated proteins like repellents and hydrophobins[35,36]. We speculate that when Rsp3-HA is expressed by *U. maydis*, the protein could have a better chance to become incorporated in the cell wall already during secretion leading to the more uniform distribution. The fact that we did not detect binding to cells of *C. graminicola* and *S. cerevisiae* could either suggest that these fungi lack the substrate bound by Rsp3. Alternatively, the substrate may not be accessible. Future studies need to identify the substrate of Rsp3 and elucidate the mechanism of cell wall attachment.

Maize plants respond to *U. maydis* infection by upregulating the expression of a large set of defense-associated genes[37]. We now show that this set includes the genes for the two DUF26-domain proteins AFP1 and AFP2, which are predicted to be secreted and detected in apoplastic fluid of infected tissue. For both proteins, we were able to demonstrate that they interact with Rsp3. In the subsequent more detailed analysis of AFP1, we have shown that AFP1 is mannose binding and displays antifungal activity that requires its mannose-binding property. This is similar to the *G. biloba* protein Gnk2, which has a single DUF26 domain and has antifungal activity depending on mannose-binding[32]. Whether ginkgo trees use the antifungal activity of GNK2 to defend themselves against fungal attackers is unknown. However, the finding that Gnk2 is expressed in seeds might point to such a function. Genes encoding secreted DUF26 domain-containing proteins were also upregulated upon *M. oryzae*

 

infection in rice[38]. Our finding that AFP1 and AFP2 are strongly induced when *U. maydis* grows on the leaf surface of maize, could suggest that they are PAMP induced. As mannose binding and antifungal activity appear coupled, we consider that AFP1-binding to mannose in the fungal cell wall might affect the integrity of the fungal cell wall and thus lead to cell death. Alternatively, AFP1 and AFP2 might bind to mannosylated proteins, which are important for virulence. It has been demonstrated that the *O*-mannosylation pathway of *U. maydis* has a critical role in virulence and mutants lacking the mannosyltransferase Pmt4 are defective in early infection-related development[39]. In addition, to explain the requirement of Pmt4 also at later stages of biotrophic development, it has been hypothesized that mannosylated effectors required at a later stage of the infection, might exist[40]. The mechanism of AFP action is unknown and will require the identification of binding partners in the fungal cell wall. At present, we do not know which properties allow Rsp3 to interact with AFP1 and AFP2 and speculate that electrostatic interactions might be involved. Rsp3 has an isoelectric point of 3.8 while AFP1 has an isoelectric points of 8.0. Length of the repetitive domain and charge of Rsp3 might thus correlate with the amount of trapped AFP. The highly divergent C-terminal domains of UmRsp3 and SrRsp3 both contain several copies of the sequence motifs PG[T/Q]Q[P/N]G, P[N/K]GD[D/N]S[S/D][E/A], [G/A][D/P]E[P/S][V/L]PYG[D/A], and G[D/K]K[P/N][K/P][D/S/N][S/G]D[K/N]. As *S. reilianum rsp3* can complement the *rsp3* mutant of *U. maydis*, these motifs could constitute specific binding sites for AFP proteins. However, it is also apparent that UmRsp3 and SrRsp3 have evolved differently despite the fact that *U. maydis* and *S. reilianum* parasitize the same host. Repeat structures are extremely unstable and are prone to repeat polymorphisms[41], which we also observe in the natural *U. maydis rsp3* alleles. In addition, point mutations that may have occurred in an individual repeats could have spread throughout the repetitive C terminus via gene conversion and/or recombination events leading to scrambling and re-assortment of certain repeats. The fact that *U. maydis* and *S. reilianum* do not productively cross, may have led to fixation of the distinctly different repeat arrangements in the two species that allows the detection of only four degenerate conserved motifs, which are interspersed with motifs unique to each species (Fig. 1a). A closer look at the all unique motifs shows that short submotifs like GDA, QQP, and PYG (Fig. 1a) are actually also found in the longer degenerate motifs detected in both smut species. It will be a future task to define the actual AFP1 and AFP2 binding sites in the C-terminal domain of Rsp3.

Rsp3$_{\Delta412-869}$ lacking the most C-terminal repetitive domain but retaining the domain consisting of the degenerate [D/N]NN and PG[T/Q]Q[P/N]G repeats present in *U. maydis* and *S. reilianum* interacts poorly with AFP1 in vitro but partially complements the virulence phenotype of the *rsp3* mutant. This truncated protein still binds to fungal hyphae. To explain the partial biological activity of this truncated protein, we speculate that the attachment of this protein to the hyphal surface might hinder access of AFP1 to mannose residues in the cell wall and provide some degree of protection that is independent from the proposed specific interaction with the C-terminal domain. The finding of strong length variations in natural *rsp3* alleles could reflect evolutionary pressure to adjust to specific maize varieties that can all be colonized by *U. maydis* but which may differ with respect to AFP proteins and how much they express. In this context, it will be interesting to analyze whether *U. maydis* strains harboring long and short *rsp3* alleles vary in virulence on different races of maize or teosinte, the only other host of *U. maydis*.

It is also possible that Rsp3 has additional functions and might bind to extracellular domains of membrane-bound DUF26

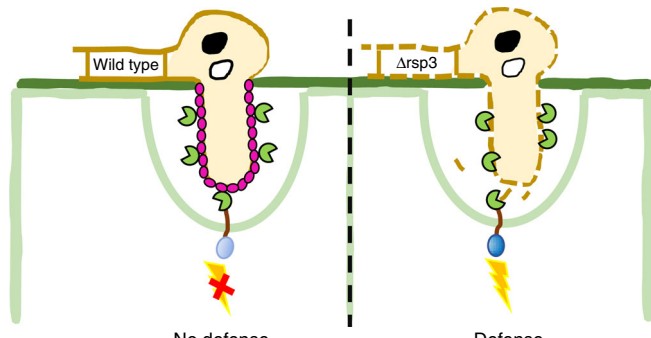

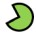 **Rsp3**

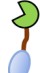 Fungal cell wall components containing mannose residues/mannosylated proteins

Maize secreted DUF26 domain containing proteins AFP1 and AFP2

Maize mannose binding membrane-bound receptor kinases (containing DUF26 domain)

**Fig. 8** Hypothetical model for the function of the *U. maydis* effector Rsp3. In response to infection by *U. maydis*, maize antifungal genes AFP1 and AFP2 are transcriptionally induced and AFP1 and AFP2 proteins are secreted to the apoplast. In infections with *rsp3* mutants, AFP1 and AFP2 target mannose residues/mannosylated proteins on the fungal cell wall negatively impacting fungal cell wall integrity. This results in killing of fungal cells which may release MAMP molecules that stimulate immune responses. To counteract this, *U. maydis* secretes Rsp3 protein which binds to fungal hyphae and protects hyphae against maize AFP proteins. Rsp3 might also interact with membrane-bound receptor kinases containing a DUF26 domain to block signaling and prevent maize immune responses

members known as cysteine-rich RLKs (CRKs), which have a role in plant defense responses[42–44]. Rsp3 could specifically recognize structural features of DUF26 proteins also present in the extracellular domains of such CRKs and via this block downstream defense signaling. This might additionally contribute to a successful colonization. On the basis of the results obtained in this study, our hypothesis on how the Rsp3-AFP interaction could affect *U. maydis* virulence is depicted in a schematic model (Fig. 8).

By transient silencing of AFP1 and AFP2, the virulence phenotype of the *rsp3* mutant is partially rescued. This provides a direct link between Rsp3 and AFP. As we were unable to restore virulence fully in the silenced lines, this could indicate that Rsp3 has additional functions. Alternatively, the outcome may be influenced by the crosstalk between AFP1 and AFP2, i.e., the observation that AFP2 was upregulated in some plants, which should have been silenced for AFP1 as well as AFP2. At this point, we cannot exclude that other DUF26 encoding proteins with antifungal activity may be upregulated in the silenced plants, and this could prevent full rescue of the *rsp3* virulence phenotype.

The finding that AFP1 and AFP2 silenced plants show stronger disease symptoms also of wild type *U. maydis* strains suggests that Rsp3 provides only partial protection against these antifungal maize proteins. Plants silenced for AFP1 and AFP2 also become more susceptible to the anthracnose fungus *C. graminicola* suggesting that the antifungal activity of AFP1 (and likely of AFP2 and other secreted DUF26-domain proteins) is directed against several fungal pathogens of maize.

In future, it will be important to investigate whether proteins like Rsp3 that provide protection from antifungal proteins

targeting mannose also exist in other fungi. Protection of fungal hyphae from plant defense responses has so far been ascribed mainly to LysM domain proteins, which bind chitin and protect hyphae from plant chitinases. Plant pathogenic fungi can possess several LysM domain effectors, which can reside either in the apoplast or be associated with fungal hyphae[45]. We could envision that fungi have developed a similarly broad defense spectrum against mannose-binding antifungal proteins. So far, Rsp3-related effectors have been detected only in smut fungi. The finding that *C. graminicola* becomes more virulent on maize plants silenced for AFP1 and AFP2 but lacks an effector related to Rsp3 could suggest that this fungus has developed another strategy to provide at least partial protection against the antifungal activity of these maize proteins. In the long run, it also should be interesting to investigate whether the upregulation of DUF26 domain-containing plant antifungal proteins could provide a general strategy for increasing resistance against fungal pathogens. As secreted DUF26-containing proteins are found in monocot as well as dicot plants, successful fungi might all have evolved molecules to counteract this type of defense.

## Methods

**Strains and growth conditions**. The *Escherichia coli* DH5α (Bethesda Research Laboratories) and Top10 (Invitrogen) were used for the cloning purposes. The haploid *U. maydis* strains FB1, FB2, as well as the haploid solopathogenic strains SG200 and SG200AN1 have been described[2,22,25], and all *U. maydis* strains used in this study are listed in Supplementary Table 3. *U. maydis* strains were grown at 28 °C in liquid complete medium (CM) with 1% glucose (Holliday, 1974), liquid YEPSL (0.4% yeast extract, 0.4% peptone, 2% sucrose), or on solid potato dextrose (PD) (2.4% PD broth [Difco], 2% Bacto agar [Difco]) plates. *C. graminicola* CgM2 (M1.001)[46] was maintained at 25 °C on oatmeal agar [Difco] plates with continuous exposure to daylight.

**Genomic DNA preparation and *U. maydis* transformation**. Standard molecular techniques for cloning and heterologous gene expression followed described protocols[47]. Transformation of *U. maydis* and genomic DNA isolation were performed as described previously[48–50].

**Gene expression analysis**. To analyze *rsp3* expression by qRT-PCR, 6-day-old maize plants of the variety *Early Golden Bantam* (Urban farmer Seeds Inc., Westfield, IN, USA) were syringe inoculated with a mixture of *U. maydis* strains FB1xFB2 or $H_2O$. For early time points where no disease symptoms were visible, leaf areas from the third leaf extending from the injection holes to 1 cm below were cut and corresponding areas from $H_2O$ inoculated plants were taken. For later time points, infected areas from 1 cm downstream of the injection holes displaying anthocyanin induction and tumor development were excised, and corresponding segments from $H_2O$ inoculated plants were collected. For each sample, the material of 10 plants was pooled, frozen in liquid $N_2$, and stored at −80 °C until RNA extraction. Plant material was ground to a fine powder in liquid nitrogen using a Retsch CryoMill (Retsch GmbH, Haan, Germany) with a 50 ml grinding beaker and a 20 mm grinding ball. The machine was precooled for 30 s followed by 60 s of grinding at 20 Hz. Approximately 500 mg of powder were resuspended in 1 ml TRIzol reagent (Life technologies) and total RNA extracted according to the manufacturers recommendation. The total RNA was DNase-treated and cDNA was prepared using TURBO DNA-free kit (Invitrogen Cat#AM1907) and Super-Script® III First-Strand Synthesis SuperMix (Invitrogen Cat#18080400), respectively. Quantitative real-time PCR analysis was performed as described[51] using primer pairs for indicated genes listed in Supplementary Table 2. The expression of constitutively expressed *U. maydis* peptidyl-prolyl isomerase (*ppi*) was used for normalization. Details of the RNAseq analysis have been described elsewhere[23] and all data relevant for this analysis were extracted from this data set available at NCBI Gene Expression Omnibus under accession number GSE103876. The qRT-PCR of AFP1 and AFP2 after infection analysis was performed with the same RNA samples previously used by Lanver et al.[23] and GAPDH of maize was used for gene normalization.

**Plasmid and strain constructions**. PCR reactions were performed using the Phusion® High-Fidelity DNA Polymerase (New England Biolabs). Point mutations were generated using the QuikChange Lightning Site-Directed Mutagenesis Kit (Agilent Technologies). Restriction enzymes were purchased from New England Biolabs. Gene replacements and integrations into the *ip* locus[52] were verified by Southern blot analysis. All fragments amplified by PCR were sequenced in the final plasmids.

All primer sequences used are listed in Supplementary Table 2. To generate Δ*rsp3* mutants, the 1.0 kb fragments from upstream and downstream of *rsp3* genes were amplified by PCR using primer pairs Rsp3 #1/2 and Rsp3 #3/4 using FB1 genomic DNA as template. The *Sfi*I-digested PCR fragments were ligated with a 1.9 kb *Sfi*I-digested fragment containing a hygromycin resistance cassette isolated from pBS-hhn[53]. The ligation product was PCR amplified using primer pair Rsp3 #1/4 and then cloned into pCR®2.1-TOPO® Vector (Invitrogen) to yield plasmid pCR2-rsp3KO. PCR-generated linear DNA of the insert was used for *U. maydis* transformation to yield strains SG200Δ*rsp3* and SG200AN1Δ*rsp3*.

For the complementation of Δ*rsp3* mutants, linearized plasmids were integrated into the *ip* locus of SG200Δ*rsp3* and SG200AN1Δ*rsp3*. Plasmid prsp3 was generated by inserting a PCR fragment containing about 1.5 kb native *rsp3* promoter region and the entire open reading frame (ORF) amplified by primer pair Rsp3 #5/6 from FB1 DNA into *Nde*I and *Not*I sites of plasmid p123[54]. Plasmid potef-rsp3-HA was created in two steps. *Afl*III/*Nde*I digested fragment from prsp3 plasmid containing *rsp3* ORF was ligated into *Sma*I/*Not*1 sites of p123 and filled gaps with Klenow to generate potef-rsp3. The plasmid potef-rsp3 was used as template to perform an inverse PCR with primer pair HA-F/R to generate potef-rsp3-HA. The potef-rsp3-HA was digested with *Nde*I/*Sfi*I to remove the *otef* promoter and replaced with a *Nde*I/*Sfi*I fragment containing the *rsp3* promoter excised from prsp3 to yield prsp3-HA. To generate pHA-rsp3, two PCR fragments were amplified using primer pairs Rsp3 #7/8 and Rsp3 #9/10 and inserted into *Hind*III/*Sfi*I sites of prsp3 using Gibson assembly strategy[55]. Plasmid prsp3(Δcys) was generated via the same strategy using primer pairs Rsp3 #11/12 and Rsp3 #13/14 and inserted into *Hind*III/*Ngo*MIV sites of prsp3. Plasmid prsp3(T6) was created by amplifying the ORF of T6 *rsp3* from Toluca-6 genomic DNA with primer pairs Rsp3 #15/16. The PCR product was digested with *Bam*HI and *Not*I and replaced the *rsp3*(FB1) ORF in prsp3. To generate plasmid prsp3(T6)$_{9CA}$, point mutations in cysteine residues were introduced using the QuikChange Lightning Site-Directed Mutagenesis Kit with primers Rsp3 #17–25 and prsp3(T6) as template. For potef-rsp3$_{A24-60}$-HA, two PCR fragments amplified with primer pairs Rsp3 #26/27 and Rsp3 #28/29 were combined and inserted into the *Bam*HI/*Bsu*36I linearized potef-rsp3-HA using Gibson assembly strategy. To generate potef-Myc$_{47}$-rsp3-HA, PCR products generated by primer Rsp3 #30/31 and Rsp3 #32/33 were ligated into *Bam*HI/*Ngo*MIV sites of potef-rsp3-HA. Plasmids potef-Myc$_{47}$-rsp3(RRDG*)-HA and potef:Myc$_{47}$-rsp3(FFRRD*)-HA were generated using the same strategy by Gibson assembly by combining two PCR fragments amplified using either primer pairs Rsp3 #30/34 and Rsp3 #35/36 or primer pairs Rsp3 #30/37 and Rsp3 #38/36 and inserted into the *Xba*I/*Bsu*36I sites of potef-Myc$_{47}$-rsp3-HA. To generate pMyc$_{47}$-rsp3(RRDG* or FFDDG*)-HA, same strategy was used to replace *otef* promoter of potef-Myc$_{47}$-rsp3(RRDG* or FFDDG*)-HA with a *rsp3* promoter as described in generating prsp3-HA. Plasmid potef-rsp3$_{9CA}$-HA was generated via QuikChange Site-Directed Mutagenesis kit using same primers for prsp3(T6)$_{9CA}$ and potef-rsp3-HA as template. To generate potef-rsp3$_{Δ412-869}$-HA, a PCR product was amplified using primer pair Rsp3 #30/39 and potef-rsp3-HA as template. The PCR product was digested with *Bam*HI and *Not*I and cloned into p123. Otef promoter was replaced with *rsp3* promoter to yield prsp3$_{Δ412-869}$-HA using same strategy as described for generating prsp3-HA. Plasmid potef-rsp3$_{Um-Sr}$-HA was generated using the same strategy via Gibson assembly by combining two PCR fragments amplified using primer pair Rsp3 #40/41 and potef-rsp3-HA as template and primer pair Rsp3 #42/43 and *S. reilianum* gDNA as template and inserted into the *Bsp*EI/*Not*I sites of potef-rsp3-HA. Plasmid pSr-rsp3-HA was generated by combining two PCR fragments amplified using primer pair Rsp3 #44/45 and *S. reilianum* gDNA as template and primer pair Rsp3 #46/47 and potef-rsp3$_{Um-Sr}$-HA as template and inserted into the *Bam*HI/*Not*I sites of potef-rsp3-HA.

**Protein secretion assay**. Total protein fractions from cell pellets and culture supernatants of *U. maydis* were prepared as described previously[7]. Briefly, *U. maydis* cells were grown in CM medium to an OD$_{600}$ of 0.6. Cell cultures were concentrated and adjusted to an OD$_{600}$ of 20 with 1× sample buffer (50 mM Tris-HCl, 2% SDS, 10% glycerol, 100 mM DTT, 0.01% bromophenol blue, pH 6.8). Glass beads were then added to the samples prior to cell disruption using FastPrep-24 homogenizer (MP Biochemicals). Proteins from supernatants were TCA-precipitated and acetone-washed twice before dissolved in sample buffer. Proteins from cell pellets and supernatant fractions were separated by SDS-PAGE. Proteins were detected in Western blot analyses using mouse monoclonal anti-HA (1:10,000 dilution, Sigma Cat#H9658), anti c-Myc (1:10,000 dilution, Sigma Cat#M4439), anti-tubulin (1:2000 dilution, Merck Cat#CP06), or rabbit anti-mCherry (1:3000 dilution, BioVision Cat#5993) antibodies, and horseradish peroxidase-conjugated anti-mouse or anti-rabbit IgG (1:10,000 dilution, Cell Signaling) as secondary antibody.

**The cleavage site of Rsp3 by N-terminal protein sequencing**. SG200Δ*rsp3*-P$_{otef}$-rsp3-HA cells were grown in CM liquid medium to an OD$_{600}$ of 0.6, the supernatant fraction was collected and TCA-precipitated overnight at 4 °C. The TCA-precipitated pellet was acetone-washed twice and dissolved in RIPA buffer (50 mM Tris, 150 mM NaCl, 0.1% SDS, 1.0% IGEPAL® CA-630, 0.5% sodium deoxycholate, pH 8.0). Rsp3-HA was immunoprecipitated using anti-HA agarose beads (Roche Cat#11815016001). The bound Rsp3-HA on the beads was boiled in

1× sample buffer and separated by SDS-PAGE. Following blotting to a PVDF membrane and staining with coomassie brilliant blue R-250, the 150 kD band corresponding to Rsp3-HA was excised from the membrane and sent for N-terminal protein sequencing by Edman degradation (TopLab, Munich, Germany).

**Appressorial quantification, penetration, and virulence**. The in vitro induction of filaments and appressoria of *U. maydis* was performed as described[25] using strains that expressed an appressorial marker (AM1) fused to GFP[24]. To determine appressoria formation and penetration efficiency of hyphae on the leaf surface established protocols were followed[24,25]. Briefly, infected maize leaves were harvested 18 h after inoculation with a strain expressing the appressorial marker and stained with Calcofluor white. To quantify appressoria, the percentage of filaments showing eGFP fluorescence was determined relative to filaments stained with Calcofluor white. To determine penetration efficiency of hyphae, the number of appressoria that have penetrated the epidermis was determined by visualizing eGFP fluorescence of intracellular hyphae relative to the total number of appressoria showing eGFP fluorescence.

Plant infections were performed in a glasshouse as described and disease severity was scored at 12 dpi following the disease rating established previously[2]. In general, about 40 plants were infected by the same strain in a single experiment and results from three independent infections done in the same season were combined in the respective graphs depicted. Because of seasonal differences in amounts of additional sunlight, disease severity can vary in experiments done at different times in the year. To take this into account, the wild type SG200 and the *rsp3* mutant were included in all assays to allow direct comparisons with strains to be tested for virulence. As a consequence, the virulence of strains tested in different experiments, i.e., figures, cannot be directly compared and has to be assessed in relation to the control strains tested at the same time.

**Protein purification**. AFP1 protein was overexpressed in *N. benthamiana* using the TMV-based viral provector system as described[56] and kindly provided by NOMAD. For the generation of plasmid pICH-AFP1-His in which AFP1-His is expressed including its own signal peptide, one PCR fragment was amplified using primer pair AFP1 #1/2 with cDNA template from SG200-infected plants harvested 3 dpi and the other two fragments were amplified from the 3′provector pICH31070 as template, using primer pairs AFP1 #3/4 and AFP1 #5/6, respectively. The three fragments were combined and inserted into the *PmeI/PspOMI* sites of pICH31070 using Gibson assembly strategy. Same strategy was used to create pICH-AFP2-His using primer pairs AFP2#1/2, AFP1#3/AFP2#3, and AFP2#4/AFP1#6. To generate pICH-AFP1*-His, three amino acid substitutions in the N-terminal mannose binding residues were introduced using QuikChange Site-Directed Mutagenesis Kit and primers AFP1#7-9 and pICH-AFP1-His as template. Plasmid pICH-AFP1**-His was generated with the same strategy using primers AFP1#10-12 and pICH-AFP1*-His as template. The plasmids and the empty vector 5′provector (pICH21011) were delivered by *Agrobacterium tumefaciens* strain GV3101 into *N. benthamiana* as described[6]. Leaf samples harvested at 4 dpi were immersed in the infiltration buffer (50 mM Tris-HCl, 50 mM CaCl$_2$, and 150 mM NaCl, pH 7.5), subjected to vacuum infiltration and centrifuged to obtain apoplastic fluid as described[57]. The apoplastic fluid was about twofold concentrated using 10 kD cutoff centrifugal filters (Millipore Inc.). Overall, 5 ml of apoplastic fluid was mixed with 20 ml of binding buffer (25 mM Tris-HCl, 150 mM NaCl, and 15 mM imidazole, pH 7.5) for application to NTA-agarose beads. The beads were washed with binding buffer and AFP1-His was eluted using the binding buffer-containing 150 mM imidazole. Protein concentrations were determined using NanoDrop spectrophotometer (Thermo Scientific).

**Mannose-binding assay**. One μg of AFP1-His wild type or mutant versions or 2 μg of Cmu1-His was added to 30 μl mannose-agarose beads (Sigma Cat#M6400) in 1 ml of binding buffer (10 mM Tris-HCl, 20 mM CaCl$_2$, 1 M NaCl, and 0.05% Tween-20, pH 7.5). After 1 h incubation at 4 °C on a rotating wheel, the beads were washed twice with binding buffer. Beads with bound proteins were boiled and subjected to western blotting analysis using anti-His antibodies conjugated to horseradish peroxidase (HRP) (1:8000 dilution, Qiagen Cat#34460).

**Growth inhibition assay**. *U. maydis* strains were grown to OD$_{600}$ = 0.8 and diluted to OD$_{600}$ = 0.001 with 10 mM Tris-HCl (pH 7.0). The AFP proteins were added to the cells to a final concentration of 1 μg/μl in a 50 μl final volume. The cells were incubated at 28 °C for 3 h before spotted on a PD agar plate. The plate was incubated on 28 °C for 1–2 days. To directly visualize cell death, AFP1-treated cells were stained with 0.1 μM SYTOX orange dye (Molecular Probes Cat#11368) for 5–10 min before visualization with epifluorescence DIC microscopy.

**Identification of Rsp3-interacting proteins**. *U. maydis* SG200Δrsp3-P$_{otef}$-rsp3-HA was grown in 10 ml of CM liquid medium to an OD$_{600}$ of 0.6. The supernatant fraction was collected, concentrated and buffer exchanged with 25 mM Tris-HCl (pH 7.5) using 30 kD cutoff centrifugal filters (Millipore). The supernatant was applied to anti-HA affinity matrix (Sigma Cat#11815016001) and washed twice with a binding buffer (25 mM Tris-HCl, 0.3 M NaCl, and 0.1% NP-40, pH 7.5). To collect apoplastic fluid from infected maize leaves, the leaves harvested at 3 dpi

were immersed in ice-cold buffer (0.2 M CaCl$_2$ and 5 mM NaAcetate, pH 4.3) for 1 h before vacuum-centrifugation method was applied as described[57]. The apoplastic fluid was concentrated and exchanged with the binding buffer before incubated with the anti-HA affinity matrix-containing Rsp3-HA. After overnight incubation at 4 °C, the bound proteins to Rsp3-HA on beads were boiled and subjected to SDS-PAGE and stained using SilverQuest Silver Staining Kit (Invitrogen Cat#LC6070). The band of interest was excised and followed by protein extraction and digestion. The peptides were analyzed using MALDI-TOF/TOF (4800 ABI-SCIEX) for analysis.

**Pull-down assay of purified Rsp3-HA with AFP-His proteins**. AFP1-His and AFP2-His expressed and purified from *N. benthamiana* were coupled to NTA-agarose beads in binding buffer (25 mM Tris-HCl, 0.3 M NaCl, and 0.1% NP-40, pH 7.5). The beads were then incubated with the supernatant collected from SG200Δrsp3-derived strains constitutively expressing secreted Rsp3-HA variants for 4 h at 4 °C. The beads were washed with binding buffer and proteins bound were removed by boiling in sample buffer and subjected to western blotting analysis using anti-His-HRP (1:8000 dilution, Qiagen Cat#34460) and mouse monoclonal anti-HA antibody (1:10,000 dilution; Sigma Cat#H9658). Horseradish peroxidase-conjugated anti-mouse or IgG (1:10,000 dilution; Sigma Cat#H9658) was then used as secondary antibody.

**FoMV-induced gene silencing**. The SGN VIGS tool (http://vigs.solgenomics.net/) was used to select an appropriate region for designing the silencing gene constructs for AFP1 and AFP2 genes. The 300 bp gene fragments of AFP1 and AFP2 were amplified by primer pairs antiAFP1-F/R and antiAFP2-F/R, respectively, and cloned into *XbaI/XhoI* sites of the Foxtail mosaic virus pFoMV-V plasmid in an antisense orientation as described[34] yielding plasmids pFoMV-AFP1 and pFoMV-AFP2, respectively. Six-day-old-maize seedlings were inoculated with either pFoMV-V, pFoMV-AFP1 pFoMV-AFP2 by biolistic particle delivery system (BioRad). The viral sap was collected 10 days post bombardment by grinding in phosphate-buffered saline (1×PBS, pH 7.4). Six-day-old seedlings were rub-inoculated using Parafilm M. The Parafilm M was placed on top of wetted paper towels inside square petri dishes and incubated at 28 °C for 15 h[25]. The parafilm were washed with PBS before 3% BSA blocking and then incubated in PBS buffer-containing α-HA antibody (Sigma; 1:1.500 dilution) and 3% BSA at 4 °C overnight. The samples were washed with PBS and incubated in the goat anti-mouse IgG secondary antibody conjugated with Alexa Fluor 488 (Life Technologies; 1:1.500 dilution) for 1 h at room temperature. After washing, the samples were analyzed using the confocal laser fluorescence microscope SP5 (Leica, Bensheim).

For immunostaining Rsp3-HA in biotrophic hyphae, infected leaves were harvested at 3 dpi and the epidermal layers of leaves were peeled off. Leaves were treated for 90 min in cell wall macerating solution (10 mM MES pH 5.7, 1.5% cellulase from *Trichoderma* sp., 0.3 % macerozyme R10, 0.6 M mannitol, 1 mM CaCl$_2$, and 0.1% BSA), and then washed with PBS and fixed in 4% paraformaldehyde for 30 min at room temperature. After washed with PBS buffer, the leaves was incubated with 0.1 M glycine in PBS for 15 min. Visualization followed the protocol described for hyphae on Parafilm M.

Transmission electron microscopy and immunogold labeling of HA was performed as described[8]. A 15 nm gold-conjugated secondary antibody (British BioCell International, Cardiff, UK) was used in this study. A JEOL JEM-1010 transmission electron microscope equipped with a XR-16 CCD camera (Advanced Microscopy Techniques, AMT, Woburn, MA, USA) was used to image the samples. The obtained data were analyzed with SPSS Statistics (IBM Corp. New York, USA) by applying the Mann–Whitney *U*-test.

**Immunolocalization of Rsp3-HA**. To localize Rsp3-HA in budding cells and filamentous hyphae, *U. maydis* strains expressing Rsp3-HA constitutively were suspended in 2% YEPSL containing 0.1 mM 16-hydroxy hexadecanoic acid at a final OD of 0.5 and sprayed on top of the Parafilm M. The Parafilm M was placed on top of wetted paper towels inside square petri dishes and incubated at 28 °C for 15 h[25]. The parafilm were washed with PBS before 3% BSA blocking and then incubated in PBS buffer-containing α-HA antibody (Sigma; 1:1.500 dilution) and 3% BSA at 4 °C overnight. The samples were washed with PBS and incubated in the goat anti-mouse IgG secondary antibody conjugated with Alexa Fluor 488 (Life Technologies; 1:1.500 dilution) for 1 h at room temperature. After washing, the samples were analyzed using the confocal laser fluorescence microscope SP5 (Leica, Bensheim).

**Binding of purified Rsp3-HA to fungal surfaces**. The supernatant fraction of SG200Δrsp3-P$_{\text{otef}}$-rsp3-HA cells grown in 100 ml CM liquid medium to an OD$_{600}$ of 0.6 was collected, 100-fold concentrated and buffer exchanged with PBS (pH 7.4) using 30 kD cutoff centrifugal filters (Millipore). The supernatant was applied to anti-HA affinity matrix (Sigma Cat#11815016001), beads were washed three times with PBS buffer, and Rsp3-HA was eluted with 0.1 M glycine (pH2.5). The eluate was neutralized with 1 M Tris-Cl (pH 8), concentrated and buffer exchanged with PBS buffer using 3 kD cutoff centrifugal filters. The purified Rsp3-HA was added to *U. maydis* SG200 (OD$_{600}$ = 0.1) budding cells or SG200 hyphae induced to filament by spraying on parafilm[25], *C. graminicola* CgM2 (10$^5$ conidia/ml), or *S. cerevisiae* AH109 (OD$_{600}$ = 0.1) cells in a final concentration of 0.01 μg/μl. After overnight incubation at 4 °C, cells were harvested and washed with PBS buffer twice before performing the immunolocalization.

**Microscopy**. A Zeiss Axioplan II microscope with differential interference contrast optics was used for microscopy. The infected plant samples were treated as described[58]. The samples were stained with WGA-AF488 (Molecular Probes, Karlsruhe, Germany) and Propidium Iodide (Sigma) to visualize fungal hyphae and plant cell wall, respectively. Confocal microscopy was performed using a TCS-SP5 confocal microscope (Leica Microsystems). Propidium iodide fluorescence was excited at 561 nm and detected at 580–630 nm. WGA-AF488 was excited at 488 nm and subsequent detected at 500–540 nm. mCherry was excited at 561 nm and detected at 580–630 nm. Images were processed using LAS-AF software (Leica Microsystems).

**Bioinformatic analyses**. Signal peptide prediction was performed with the program SignalP 4.1 (http://www.cbs.dtu.dk/services/SignalP/)[26]. Sequence alignments were generated using Clustal Omega (http://www.ebi.ac.uk/Tools/msa/clustalo/)[59]. Domain analyses were performed with Pfam (http://pfam.xfam.org/)[60]. Protein modeling was performed with Swiss-Model (https://swissmodel.expasy.org/interactive)[61] and images were superimposed using PyMOL (https://www.pymol.org/).

**Data availability**. The RNAseq data were previously reported[23], and are available at NCBI Gene Expression Omnibus under accession number GSE103876. The data generated for this study are available in this article and the Supplementary Information Files. Other data that support the findings of this study are available from the corresponding author upon request.

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

## Acknowledgements

We thank O. Paredes-Lopez from the Center for Research and Advanced Studies of the National Polytechnic Institute in Irapuato for providing *U. maydis* strains isolated from different locations in Mexico. We are grateful to Nomad Bioscience GmbH for providing the magnICON® viral pro-vectors for protein expression in *N. benthamiana,* Holger B. Deising for making available *C. graminicola* M2 and Steven A. Whitham for providing the Foxtail mosaic virus system for gene silencing. We thank Libera Lo Presti for SG200 P_{cmu1}-mCherry-AvitagHA, Gabriel Schweizer for bioinformatics support, Xiaowei Han for providing Cmu1-His protein, André Müller for sharing RNA samples for the qRT-PCR analysis, and Liang Liang for preparing samples for transmission electron microscopy. We acknowledge technical assistance of Karin Münch. We thank Stephan Wawra for comments on an early version of the manuscript. Our work was supported by funds from the Max Planck Society.

## Author contributions

L.-S.M. conceived and performed experiments. L.-S.M. and R.K. wrote the manuscript. A.M.-M., S.U., L.W., M.M., and C.T. generated mutants, identified length polymorphisms, and performed initial phenotypic analyses. A.C. did the immunostaining in planta. J.K. did the mass spectrometry analysis. B.Z. performed transmission electron microscopy. G.B. assisted in protein modeling analysis. S.R. assisted in confocal microscopy analysis. All authors discussed the results and commented on the manuscript.

## Additional information

**Competing interests:** The authors declare no competing interests.

