## [Peer Review File · Nature Communications]

Reviewers' comments:

Reviewer #1 (Remarks to the Author):

In this manuscript the authors have identified and characterized a new effector protein, Rsp3, from the phytopathogenic fungus *Ustilago maydis*. They have shown that this effector is expressed at very early time points of the interaction and its deletion causes attenuation of the virulence, with reduced tumor and anthocyanin formation. They also confirmed that the protein is secreted and processed and that it likely locates at the surface of the fungal hyphae where it possibly contributes to avoid the elicitation of plant defense responses by preventing the recognition by two maize mannose binding proteins. In general the work is sound, well carried out and uncovers a novel mechanism of virulence for this fungus. I have though several comments and experimental suggestions that might help to strength the conclusions of the manuscript.

1. In my opinion, while it is clear from Figure 2 that deletion of Rsp3 reduces the virulence of the SG200 strain, I think the way the results are shown, only allows a narrow window of observation of the pathogenic development. It might happen, and the authors even suggest something similar in the text, that actually the colonization and pathogenicity is not impaired but delayed. Same is true for the cellular colonization shown in Figure 2d. It would show a more dynamic view of the infection to extend the analysis of the symptoms and colonization in a time course manner.

2. The analysis of the protein processing is confusing. After reading the whole paragraph and following the description of the figures, one does not know exactly what the final message of this analysis is. What is the form of the protein that the authors consider the functional form? What are the key experiments that show that? I have to say, I was lost midway and although I recognize the amount of work behind, I am not sure how relevant for the functional analysis these results are. I would consider describing them in a different way taking into consideration what the final message about the processing means (like for the interaction in later experiments with AFP1 or the localization of the protein).

3. The authors show using different experimental set ups and controls to show that Rsp3 likely binds to the surface of biotrophic hyphae in contrast to a shorter version of the protein which is not secreted and also in contrast to another effector known to enter the plant cell. They conclude that this effector binds the cell wall of the fungus during the biotrophic phase. However, this claim should be moderated since although Rsp3 is clearly secreted and stays in the plant-fungal interaface, the localization at the cell wall or plasma membrane or other any other locations is not totally elucidated. In addition, even if the effector is expressed during the biotrophic phase, from the results of the Supplementary Figure, it looks like Rsp3 binding is not specific of the biotrophic phase (meaning it does not require specific components only produced in that phase). I think this is important to highlight in the main text. In this line, and using the Rsp3-HA purified proteins that the authors use in Figure 5 it would be useful to know whether Rsp3 might also bind the surface of other phytopathogenic (or not) fungi, to extend the importance of the mechanism to other interactions (see below).

4. Two maize proteins (AFP1 and AFP2) were identified as putative interacting partners of Rsp3. While the interaction with AFP1 was further confirmed using pull down experiments, the authors do not mention anything regarding AFP2, did they also try, did it work? In this experiment, which is likely to be difficult to demonstrate through other different methods (Y2H, BiFC), I am missing a negative control that demonstrates the specificity of the interaction. For example, other AFP proteins from the same class 5, some of those, if I understand the Suppl. Figure 7d correctly, might be also induced by *Ustilago* infection, even if with lower levels of expression in general.

5. The antifungal activity of AFP1 is a key experiment. However, I think this should be more thoroughly investigated. The authors show that AFP1 is able to reduce the number of colonies formed by the strain SG200 in which Rsp3 was deleted (Figure 6c). This should be actually compared with the SG200 strain in which Rsp3 is still present, to demonstrate that in that case AFP1 does not have, or not so much, antifungal activity. This experiment would be even more convincing if done with titration in which different amounts of recombinant AFP1 and Rsp3, and mannose are added. This also concerns the experiment shown in Figure 6d.

In this line, the experiment in Figure 7 should include a strain of SG200 in silenced plants and perhaps of other pathogenic fungi, to evaluate the general effect of silencing AFP1/2 in the susceptibility of maize to fungal infections. It might be useful to use the lines in which actually an increase in the expression of AFP2 was shown to be up-regulated.

Reviewer #2 (Remarks to the Author):

Ma et al. have characterized a novel effector of the maize smut fungus *Ustilago maydis* called Rsp3. They have shown that this protein, after secretion, associates with the smut cell wall and protects it against two antifungal apoplastic proteins of maize, AFP1 and AFP2. The finding is important and the methodology used, e.g. targeted deletion, introduction of truncations, exchange of individual amino acids and employment of virus-mediated gene silencing, represents up-to-date technology. As the manuscript significantly improves understanding of effector function and molecular interactions of a fungal pathogen with its host, I am very positive about this work. However, some additional experiments, as specified below, may help to improve the paper.

Specific comments

Line 43: In the abstract, the effector designated *rsp3* is introduced, but the reader has to wait until line 90 in order to learn that the abbreviation *rsp3* stands for Repetitive Secreted Protein 3. The explanation should be introduced at first mention.

Line 71: ... induce anthocyanin formation, ...

Line 103: ... consists ...

Paragraph starting at line 111, and Fig. 1b: The authors show *rsp3* transcript abundance data in axenic culture and in infection assays, ranging from 0.5 to 12 days post inoculation of maize plants with *U. maydis*. Are values at the Y-axis relative values? Please specify.

Line 119: Which promoter? Please specify.

Lines 126 onwards, Fig. 2: Frequently, mutagenesis does not only cause a reduction of virulence, but often results in more prominent variability of disease severity. The sub-figures shown in this manuscript do not reflect this. Standard deviations are not shown, and the reader is not informed about the reproducibility of the infection assay. Although this is a standard used in the *Ustilago* community, the authors may consider if the way the data are presented in this figure and elsewhere in the manuscript can be altered.

Lines 138-140: Sentence unclear.

Lines 141 onward: The micrographs clearly show the phenotype of the *rsp3* deletion mutants. The authors should comment on the number of independent mutants used to perform these experiments. This is of particular importance, as the complementation mutant (line 146) shows an intermediate phenotype.

Lines 158 onward: The anomalous migration rates of Rsp3-HA leave the reader somewhat helpless. The possible explanation provided in the Discussion section (lines 336 on) could be given here to cause less irritation.

Lines 162 - 164: The sentence should be modified: To better understand in which compartment the

processing occurs, we generated SG200 Δ rsp3 strains expressing mCherry47-Rsp3 (mCherry is inserted at amino acid position 47 of Rsp3) under control of the constitutive otef promoter or its native promoter.

Line 170 - 172: The authors should explain or speculate on the patchy mCherry fluorescence pattern.

Line 181: Has Myc47-Rsp3(FFRRD*)-HA been tested as well (Fig. 3)? Show data.

Line 196: ... cleaved off, as revealed by the failure ...

Line 201: ... strongly support the idea that correct processing ...

Lines 221 onwards and line 234. The strong fluorescence on the surface of budding cells is described, with reference to Suppl. Fig. 5a, and later the authors state that Rsp3 is exclusively attached to the surface of biotrophic hyphae. This is not surprising, given the expression profile shown in Fig. 1b.

The authors may want to consider performing a carbohydrate array experiment (see de Jonge et al. 2010, Science 329: 953-955). This also holds true for the maize proteins AFP1 and AFP2. These data would significantly improve understanding of carbohydrate binding specificity of the proteins investigated. The pull-down assays with mannose-conjugated sepharose are fine to show that indeed mannose is bound, but they do not say much about the specificity (see below).

Lines 247-249: The authors show direct interaction between Rsp3 and AFP1, but not with AFP2. Do these data exist? If so, please show or argue.

Lines 286 and Fig. 6: As above, the pull-down assays are fine but do not address the specificity of the carbohydrate. In Fig. 6c the authors should show dilutions (line 301). The reader wonders why AFP2 has not been investigated.

Line 296 onward: Using *U. maydis* yeast cells, the authors show that AFP1 has antifungal properties. This is only distantly related to the infection scenario, where biotrophic hyphae - not sporidia - are protected by Rps3. Using hyphae - preferentially biotrophic hyphae, if technically possible - would significantly improve the data.

Line 321: This should read: ... than plants that had received the empty ...

Lines 328 onward: I enjoyed reading the Discussion section, as it provides explanations helping to understand the data shown. There is no need for changes.

Lines 460 onward: The Reference section needs a lot of attention. In reference 2 the name Kämper is spelled incorrectly. The authors should have noticed this. The journal names are only occasionally written with capital letters. It looks as if the program used needs some attention.

In summary, I strongly suggest to publish the manuscript after thorough revision.

Response to the Referees:

Reviewer#1:

In this manuscript the authors have identified and characterized a new effector protein, Rsp3, from the phytopathogenic fungus *Ustilago maydis*. They have shown that this effector is expressed at very early time points of the interaction and its deletion causes attenuation of the virulence, with reduced tumor and anthocyanin formation. They also confirmed that the protein is secreted and processed and that it likely locates at the surface of the fungal hyphae where it possibly contributes to avoid the elicitation of plant defense responses by preventing the recognition by two maize mannose binding proteins. In general the work is sound, well carried out and uncovers a novel mechanism of virulence for this fungus. I have though several comments and experimental suggestions that might help to strength the conclusions of the manuscript.

1. In my opinion, while it is clear from Figure 2 that deletion of Rsp3 reduces the virulence of the SG200 strain, I think the way the results are shown, only allows a narrow window of observation of the pathogenic development. It might happen, and the authors even suggest something similar in the text, that actually the colonization and pathogenicity is not impaired but delayed. Same is true for the cellular colonization shown in Figure 2d. It would show a more dynamic view of the infection to extend the analysis of the symptoms and colonization in a time course manner.

We now include the colonization pattern of the *rsp3* deletion mutant at different stages in Supplementary Fig. 2c and wild type for comparison. This shows that the *rsp3* deletion mutant has a defect in proliferation and fails to spread along plant vascular bundle sheets even at later time points.

2. The analysis of the protein processing is confusing. After reading the whole paragraph and following the description of the figures, one does not know exactly what the final message of this analysis is. What is the form of the protein that the authors consider the functional form? What are the key experiments that show that? I have to say, I was lost midway and although I recognize the amount of work behind, I am not sure how relevant for the functional analysis these results are. I would consider describing them in a different way taking into consideration what the final message about the processing means (like for the interaction in later experiments with AFP1 or the localization of the protein).

We have deleted most of the data and just show that N-terminal processing does occur and where. In addition, we include the mutant lacking the region between signal peptide and processing site ($\Delta 24-60$) and show that the respective protein is no longer secreted. The chapter was completely re-written and Fig. 3d was removed.

3. The authors show using different experimental set ups and controls to show that Rsp3 likely binds to the surface of biotrophic hyphae in contrast to a shorter version of the protein which is not secreted and also in contrast to another effector known to enter the plant cell. They conclude that this effector binds the cell wall of the fungus during the biotrophic phase. However, this claim should be moderated since although Rsp3 is clearly secreted and stays in the plant-fungal interaface, the localization at the cell wall or plasma membrane or other any other locations is not totally elucidated. In addition, even if the effector is expressed during the biotrophic phase, from the results of the Supplementary Figure, it looks like Rsp3 binding

is not specific of the biotrophic phase (meaning it does not require specific components only produced in that phase). I think this is important to highlight in the main text. In this line, and using the Rsp3-HA purified proteins that the authors use in Figure 5 it would be useful to know whether Rsp3 might also bind the surface of other phytopathogenic (or not) fungi, to extend the importance of the mechanism to other interactions (see below).

We have now tested the binding of purified Rsp3-HA to *U. maydis*, *C. graminicola*, and *S. cerevisiae* and added two figures (Supplementary fig. 5 and 6). We could detect binding of Rsp3-HA only to *U. maydis* and in particular to the pole of budding cells, hyphal tips and septa. This pattern is different from what we observe when the protein is expressed by *U. maydis*. We refer to these differences and speculate that there may be differences in accessibility in the discussion.

4. Two maize proteins (AFP1 and AFP2) were identified as putative interacting partners of Rsp3. While the interaction with AFP1 was further confirmed using pull down experiments, the authors do not mention anything regarding AFP2, did they also try, did it work? In this experiment, which is likely to be difficult to demonstrate through other different methods (Y2H, BiFC), I am missing a negative control that demonstrates the specificity of the interaction. For example, other AFP proteins from the same class 5, some of those, if I understand the Suppl. Figure 7d correctly, might be also induced by Ustilago infection, even if with lower levels of expression in general.

We had included in figure 5A truncated Rsp3 protein lacking most of the repetitive domain and showed that this protein binds AFP1 protein only poorly- and we considered this to serve as a negative control. We have now performed a pull down assay to show interaction of Rsp3-HA also with AFP2-His (Supplementary Fig. 8b). We had also mentioned when we observed only partial rescue of virulence of the *rsp3* mutant in the AFP1/2 silenced maize plants that we cannot exclude that other members of secreted DUF26 domain containing proteins slightly upregulated during later phases of the infection might also have antifungal activity.

We have now added the amino acid alignments of all 10 predicted secreted maize DUF26 domain containing proteins in Supplementary Fig. 8c. In several of them are amino acids conserved which are involved in mannose-binding, i.e. these proteins could have antifungal activity. If expressed, these proteins could all act against fungal invaders.

5. The antifungal activity of AFP1 is a key experiment. However, I think this should be more thoroughly investigated. The authors show that AFP1 is able to reduce the number of colonies formed by the strain SG200 in which Rsp3 was deleted (Figure 6c). This should be actually compared with the SG200 strain in which Rsp3 is still present, to demonstrate that in that case AFP1 does not have, or not so much, antifungal activity. This experiment would be even more convincing if done with titration in which different amounts of recombinant AFP1 and Rsp3, and mannose are added. This also concerns the experiment shown in Figure 6d.

Rsp3 is not expressed at all during *U. maydis* growth in axenic culture unless we use a constitutively active promoter (P_{otef}). For this reason we cannot compare SG200 and SG200 Δ rsp3 and this is why we compared the strain SG200 Δ rsp3-Potef-rsp3-HA (constitutively expressing Rsp3) with SG200 Δ rsp3 (absence of Rsp3 protein). We have now

emphasized more strongly that wild type Rsp3 protein protects from the antifungal activity of AFP1 while mutant AFP1 unable to bind mannose, is unable to restrict growth of *U. maydis* lacking Rsp3.

Unfortunately, we were unable to purify active AFP1 protein from *E. coli*. After transient expression in *N. benthamiana* we could only purify small amounts of AFP1-His (less than 100 micrograms total per experiment). For the killing reaction we incubated 50 μ l of cells ($OD_{600}=0.001$) with AFP1-His in a final concentration of 1 μ g/ μ l and used the remaining protein for the control strain. In several independent experiments we have tested lower final concentrations of AFP1-His and found that lowering the amount down to 0.5 μ g/ μ l did not significantly reduce the *U. maydis* titer, suggesting that the antifungal activity is dose dependent. Since these results were generated with different protein preparations, we decided against including them in the manuscript but provide the data to the reviewers now.

6. In this line, the experiment in Figure 7 should include a strain of SG200 in silenced plants and perhaps of other pathogenic fungi, to evaluate the general effect of silencing AFP1/2 in the susceptibility of maize to fungal infections. It might be useful to use the lines in which actually an increase in the expression of AFP2 was shown to be up-regulated.

Unfortunately, we do not have stable silenced maize lines or overexpression lines (it would take >1 year to generate them and get enough seeds) but use lines in which the expression of AFP1/2 is transiently downregulated. There is to our knowledge currently no reliable protocol for transient overexpression of genes in maize.

We have now performed the experiment infecting FoMVsAFP1/2 plants by wild type *U. maydis* SG200 and *C. graminicola* CgM2. We show that both SG200 and CgM2 are more virulent in the silenced lines than in control lines infected with the empty vector control (Figure 7b-d). This illustrates that the antifungal activity of AFP1/2 is not restricted to *U. maydis* and in addition it shows that Rsp3 provides only partial protection against the antifungal activity. This is now also discussed.

Reviewer #2 (Remarks to the Author):

Ma et al. have characterized a novel effector of the maize smut fungus *Ustilago maydis* called Rsp3. They have shown that this protein, after secretion, associates with the smut cell wall and protects it against two antifungal apoplastic proteins of maize, AFP1 and AFP2. The finding is important and the methodology used, e.g. targeted deletion, introduction of truncations, exchange of individual amino acids and employment of virus-mediated gene silencing, represents up-to-date technology. As the manuscript significantly improves understanding of effector function and molecular interactions of a fungal pathogen with its host, I am very positive about this work. However, some additional experiments, as specified below, may help to improve the paper.

Specific comments

1. Line 43: In the abstract, the effector designated *rsp3* is introduced, but the reader has to wait until line 90 in order to learn that the abbreviation *rsp3* stands for Repetitive Secreted Protein 3. The explanation should be introduced at first mention.

This has been done.

2. Line 71: ... induce anthocyanin formation, ...

Has been changed.

3. Line 103: ... consists ...

Has been changed.

4. Paragraph starting at line 111, and Fig. 1b: The authors show *rsp3* transcript abundance data in axenic culture and in infection assays, ranging from 0.5 to 12 days post inoculation of maize plants with *U. maydis*. Are values at the Y-axis relative values? Please specify.

Yes, the Y-axis values are relative values. The constitutively expressed *U. maydis* peptidylprolyl isomerase (*ppi*) was used for normalization. *rsp3* expression in budding cells grown in axenic culture was set to 1.0. It is stated in the respective figure legend.

5. Line 119: Which promoter? Please specify.

We now specify the promoter and give the reference.

6. Lines 126 onwards, Fig. 2: Frequently, mutagenesis does not only cause a reduction of virulence, but often results in more prominent variability of disease severity. The sub-figures shown in this manuscript do not reflect this. Standard deviations are not shown, and the reader is not informed about the reproducibility of the infection assay. Although this is a standard used in the *Ustilago* community, the authors may consider if the way the data are presented in this figure and elsewhere in the manuscript can be altered.

We have now added standard deviations for all symptom categories in all figures.

7. Lines 138-140: Sentence unclear.

We have now re-written the sentence to make it more clear (lines 138-140).

8. Lines 141 onward: The micrographs clearly show the phenotype of the *rsp3* deletion mutants. The authors should comment on the number of independent mutants used to perform these experiments. This is of particular importance, as the complementation mutant (line 146) shows an intermediate phenotype.

Previously, we only showed colonization 4 and 8 days post infection by wild type, the *rsp3* deletion mutant and the *rsp3* Δ cys mutant. *rsp3* deletion mutants had already been generated in two different backgrounds (SG200 and SG200AN1). In both backgrounds the mutant showed less colonization. We now include additional pictures taken at earlier and later time points which reinforce that the *rsp3* mutant does not reach the veins where wild type hyphae accumulate starting at 2dpi (Supplementary Fig 2c). We describe these results in the results section.

9. Lines 158 onward: The anomalous migration rates of Rsp3-HA leave the reader somewhat helpless. The possible explanation provided in the Discussion section (lines 336 on) could be given here to cause less irritation.

We have now moved the explanation from the Discussion to the Results section.

10. Lines 162 - 164: The sentence should be modified: To better understand in which compartment the processing occurs, we generated SG200 Δ rsp3 strains expressing mCherry47-Rsp3 (mCherry is inserted at amino acid position 47 of Rsp3) under control of the constitutive otef promoter or its native promoter.

To avoid misunderstandings and to follow suggestions also from reviewer #1, we have now removed this paragraph as well as Fig. 3d and have completely rewritten this chapter on processing.

11. Line 170 - 172: The authors should explain or speculate on the patchy mCherry fluorescence pattern.

This paragraph including Supplementary Fig.4 have been removed in the course of streamlining this chapter on processing.

12. Line 181: Has Myc47-Rsp3(FFRRD*)-HA been tested as well (Fig. 3)? Show data.

We have now performed the infection using the strain FFRRD* and show that it can fully complement the virulence phenotype of Δ rsp3 (Supplementary Fig. 4c). Similar to the RRDG mutant protein, the FFRRD mutant protein migrates slower and two discrete protein species are detected in the supernatant fraction. This suggests that the presence of one or more additional cleavage sites between signal peptide and identified cleavage site. These data are discussed.

13. Line 196: ... cleaved off, as revealed by the failure ...

The sentence has been removed.

14. Line 201: ... strongly support the idea that correct processing ...

This has been corrected.

15. Lines 221 onwards and line 234. The strong fluorescence on the surface of budding cells is described, with reference to Suppl. Fig. 5a, and later the authors state that Rsp3 is exclusively attached to the surface of biotrophic hyphae. This is not surprising, given the expression profile shown in Fig. 1b.

The authors may want to consider performing a carbohydrate array experiment (see de Jonge et al. 2010, Science 329: 953-955). This also holds true for the maize proteins AFP1 and AFP2. These data would significantly improve understanding of carbohydrate binding specificity of the proteins investigated. The pull-down assays with mannose-conjugated sepharose are fine to show that indeed mannose is bound, but they do not say much about the specificity (see below).

We have now analyzed binding of Rsp3-HA to a glycan array to identify the substrate(s) of Rsp3. For this we used the printed array containing 300 compounds from RayBiotech, USA. We tested binding to three arrays using varying amounts of Rsp3-HA protein and antibody. Rsp3 consistently bound to monosaccharides (β -glc, β -gal, and α -man) and aminoglycosides (Tobramycin, Sisomicin, and Kanamycin). The binding to aminoglycosides might indicate that Rsp3 binds glucosamine. So far, glucosamine has not been reported to exist in the *U. maydis* cell wall. However, the analyses conducted so far have used hyphae induced by stress (low pH), which is unlikely to reflect biotrophic hyphae. Interestingly, 3 of 5 predicted genes encoding chitin deacetylase are weakly expressed during *U. maydis* growth in axenic culture and one is highly induced during biotrophic development. This could indicate that *U. maydis* deacetylates chitin to glucosamine. The gene for the upregulated chitin deacetylase has been deleted by the group of Jörg Kämper, but there was no virulence phenotype (personal communication). To follow this story up would require deleting all five chitin deacetylases simultaneously. Furthermore, since the array used does not contain the major glycan compounds present in fungal cell wall (high mannose N-glycan structures and β -1,3-glucans), we have sent Rsp3-HA protein to CFG to be tested on their glycan array containing 600 compounds. Unfortunately, we are still waiting for the results. However, we have in the meantime contacted Dr. Sabine Strahl from Heidelberg to discuss with her what would be needed to make any data derived from the arrays conclusive. It is very clear that this would require substantially more experiments and constitute a project on its own. Based on this we decided against including the very preliminary data in the manuscript.

We fully agree that it is very important to improve our understanding of carbohydrate binding proteins by determining their binding specificity in future work. For the current manuscript, we consider it out of scope to address the binding specificity of AFP1 or AFP2.

16. Lines 247-249: The authors show direct interaction between Rsp3 and AFP1, but not with AFP2. Do these data exist? If so, please show or argue.

We have now performed the pull down assay also with AFP2-His. Rsp3-HA is able to bind to AFP2-His and this data is now included in Supplementary Fig 8b.

17. Lines 286 and Fig. 6: As above, the pull-down assays are fine but do not address the specificity of the carbohydrate. In Fig. 6c the authors should show dilutions (line 301). The reader wonders why AFP2 has not been investigated.

Unfortunately, we were unable to purify active AFP1 protein from *E. coli*. After transient expression in *N. benthamiana* we could only purify small amounts of AFP1-His (less than 100 micrograms total per experiment). For the killing reaction we incubated 50 μ l of cells ($OD_{600}=0.001$) with AFP1-His in a final concentration of 1 μ g/ μ l and used the remaining protein for the control strain. In several independent experiments we have tested lower final concentrations of AFP1-His and found that lowering the amount to 0.5 μ g/ μ l did not significantly reduce the *U. maydis* titer, suggesting that the antifungal activity is dose dependent. Since these results were generated with different protein preparations, we decided against including them in the manuscript but provide the data to the reviewers now (see comments to reviewer #1).

18. Line 296 onward: Using *U. maydis* yeast cells, the authors show that AFP1 has antifungal properties. This is only distantly related to the infection scenario, where biotrophic hyphae -

not sporidia - are protected by Rps3. Using hyphae - preferentially biotrophic hyphae, if technically possible - would significantly improve the data.

Unfortunately, the only hyphae we can generate outside of the plant are hyphae which are cell-cycle arrested and hence cannot be used to assess if they are killed by the antifungal proteins. To our knowledge, it has so far not been possible to generate *U. maydis* hyphae which grow via clamp connections outside the plant and which might resemble biotrophic hyphae. And even if this were possible it is not clear if they would express all the secreted proteins that are induced only during plant colonization. Therefore, we cannot do the suggested experiments. However, by showing that virulence of the *rsp3* mutant is significantly elevated in AFP1/2 silenced plants we think that we have shown a direct connection between Rsp3 and AFP1/2 during host colonization.

19. Line 321: This should read: ... than plants that had received the empty ...

This was changed.

20. Lines 328 onward: I enjoyed reading the Discussion section, as it provides explanations helping to understand the data shown. There is no need for changes.

We are delighted to hear that you enjoyed reading this section.

21. Lines 460 onward: The Reference section needs a lot of attention. In reference 2 the name Kämper is spelled incorrectly. The authors should have noticed this. The journal names are only occasionally written with capital letters. It looks as if the program used needs some attention.

This has been carefully checked now.

In summary, I strongly suggest to publish the manuscript after thorough revision.

Thanks

Reviewers' comments:

Reviewer #1 (Remarks to the Author):

In this version of the manuscript the authors have addressed satisfactorily most of the comments raised by the referees in the previous version, and the manuscript has gained in clarity and significance. However, I think there are still some points that deserve attention. In general some of the conclusions should be toned down as several issues have not been addressed or addressed but could not be solved experimentally.

I have listed several issues in order of appearance in the manuscript, the most important are though the last three points.

- Lines 43 to 46, The gene/protein nomenclature is not clear. I understand that the effector is Rsp3 and not *rsp3* which is the gene, and thus when talking about function it should rather refer to Rsp3
- Lines 74-79, this paragraph is somehow reiterative and not fully clear. How many proteins with repeats in total, how many with Kex2 sites? Combining sentences 1 and 3 perhaps helps.
- Line 105 and corresponding introduction, it is surprising the omission of the fungal effector SP7 from *Glomus* when describing repeat-containing fungal effector proteins.
- Line 112, The expression of Rsp3 peaks at 2 dpi, and the authors conclude here that the effector is specifically required during early stages of infection. Although I agree with this statement, I wonder now knowing the putative function of Rsp3, why this is the case and why it is not required at "middle" stages. These should be discussed perhaps in lines 334 and forward.
- Lines 233 to 235, this sentence implies that these plant proteins/genes are differentially regulated upon infection. This is interesting and it should be also discussed. Is there any other evidence of this? At the gene level? During other interactions. Please discuss.
- Line 244, "AFP1-His could pull down Rsp3-HA, chimeric Rsp3Um-Sr-HA and Rsp39CA-HA (Fig. 5a-b). However, Rsp3 Δ 412-869-HA poorly interacted with AFP1-His (Fig. 5a), suggesting that the interaction with AFP1 is largely occurring through the C-terminal repetitive domain of Rsp3." I do not see this clearly in the corresponding Figure, the Rsp39CA-HA is almost as poorly pulled down Rsp3 Δ 412-869-HA. I think this figure clearly demonstrates that Rsp3 interacts with AFP1, but I do not agree with the conclusion that the carboxy terminus containing the repeats is the only region responsible for the binding, because loss of the 9 Cysteins (why was not the version without the whole Cys domain used?) also poorly pulls down Rsp3. This also makes sense in terms of virulence, because both strains are the less virulent than the wild type, supporting the hypothesis that interaction with AFC1 is critical for virulence. See below.
- Line 252, "Rsp3 Δ 412-869-HA was only partially able to complement the virulence phenotype of an *rsp3* deletion strain (Supplementary Fig. 4d)." To me, it does not complement at all if compared with the phenotype of the Δ *rsp3* shown in Figure 2. If analysed together, it can be observed that there are major differences in the number of heavy tumors and normal tumors between the two experiments employing the Δ *rsp3* strain shown in Figure 2a and Supp. 4d, being the virulence phenotype of the deletion stronger in Supp. 4d, and thus, the differences to the other strains, such as Rsp3 Δ 412-869-HA, more exacerbated. Please show all those plant phenotypes together to be able to compare. If this is the case, the strength of the conclusions should be toned down.
- Line 296, "This illustrates that AFP1 has antifungal activity and mannose binding is crucial for this activity." As mentioned in my previous review, this statement is not fully sustained without further proof, like the titration of the binding by adding mannose that should outcompete the binding to *rsp3* and thus the antifungal activity. If no other data are supplied, I understand the other difficulties the authors mention, the statements again should be moderated.

Reviewer #2 (Remarks to the Author):

The revised version of the manuscript by Ma et al., several critical points raised by the reviewers have been addressed. No need to highlight again the wealth of data and the novelty of the finding that Rsp3 allows the pathogen escaping the activity of AFP1/2. Among the improvements were the inclusion of another, rather distantly related, plant pathogenic fungus, *C. graminicola*. This added to understanding the specificity of the *Ustilago* effector Rsp3, and of the maize defense proteins AFP1 and 2. Unfortunately, the glycan binding specificity of Rsp3 has been addressed by probing a RayBiotech-300-compounds array. This approach led to inconclusive results, and a 600-compounds-array is under way. I found this irritating and was wondering why the authors did not have the patience to wait for the answer. I understand that careful binding assays might take a lot of effort. However, as the binding specificity of the effector is important in order to understand its mode of protection, one may not share the author's view that this is out of scope of this manuscript. This still is a weak point of the manuscript. Further regarding the specificity of the effector, the authors tested binding of Rsp3 to *S. cerevisiae* and *C. graminicola* cells. Given the enormous variability of carbohydrate exposition on hyphae at different stages of infection, one may question whether it is sufficient to look at binding of Rsp3 to conidia. The expression data shown indicate that the Rsp3 gene is most strongly expressed at the initial biotrophic stage. So, how does this relate to *C. graminicola* conidia or yeast cells?

So, while several critical points have been taken care of, some questions remain to be answered.

Specific comments:

Lines 104 and 106: DNA sizes should be given in a consistent way, either bp or kbp.

Line 217: ... also when added externally.

Lines 221 - 223: Comments on the binding specificity of Rsp3 are to be found above. Again, as hyphal surfaces may differ, depending on their lifestyle, and certainly differ from spore surfaces, data shown in Suppl. Fig. 6a and b may be difficult to interpret.

Line 232: Sentence is incomplete.

Line 238: The abbreviation AFP should be explained at first mention.

Lines 295 and 296, and Fig. 6c: The authors estimate titers (about 4 to 5-fold lower...). Can more precise data be provided?

Lines 302 - 304: This sentence gives redundant information (see lines 295 and 296) and should be deleted.

Fig. 7 shows infection assays. Again, standard deviations are missing in the *Ustilago* assays. The differences of *Colletotrichum* disease symptoms on FoMV as compared to FoMV_AFP1/2 leaves are obvious. I strongly suggest to use qPCR to quantify fungal DNA, and to use that as a quantitative criterion of fungal pathogenic development. Counting spots is rarely reproducible, which may be the reason why standard deviations are missing in Fig. 7d as well. As mannose-binding AFP1/2 affect the *C. graminicola* infection process and disease severity, data shown in Suppl. Fig. 6a may even be harder to interpret. Labeling cross-sectioned biotrophic and necrotrophic hyphae may lead to improved understanding of the specificity of Rsp3 and AFP1/2. Furthermore, as Rsp3 appears to play a prominent role at early biotrophic stages of infection, would the infection process of a necrotroph be altered in virulence in the presence/absence of AFP1/2?

Reviewers' comments:

Reviewer #1 (Remarks to the Author):

In this version of the manuscript the authors have addressed satisfactorily most of the comments raised by the referees in the previous version, and the manuscript has gained in clarity and significance. However, I think there are still some points that deserve attention. In general some of the conclusions should be toned down as several issues have not been addressed or addressed but could not be solved experimentally.

I have listed several issues in order of appearance in the manuscript, the most important are though the last three points.

1. Lines 43 to 46, The gene/protein nomenclature is not clear. I understand that the effector is Rsp3 and not *rsp3* which is the gene, and thus when talking about function it should rather refer to Rsp3.

The referee is right and we have changed this throughout the manuscript.

2. Lines 74-79, this paragraph is somehow reiterative and not fully clear. How many proteins with repeats in total, how many with Kex2 sites? Combining sentences 1 and 3 perhaps helps.

We have now combined sentences 1 and 3 to make this clear.

3. Line 105 and corresponding introduction, it is surprising the omission of the fungal effector SP7 from *Glomus* when describing repeat-containing fungal effector proteins.

We have now added a paragraph on repeat-containing translocated effector proteins and provide the respective references.

4. Line 112, The expression of Rsp3 peaks at 2 dpi, and the authors conclude here that the effector is specifically required during early stages of infection. Although I agree with this statement, I wonder now knowing the putative function of Rsp3, why this is the case and why it is not required at “middle” stages. These should be discussed perhaps in lines 334 and forward.

We now refer to a recent RNAseq study from our lab (Lanver et al., 2018) which is now in press which places *rsp3* in an effector-enriched module associated with the establishment of biotrophy, i.e. the early stages of fungal development inside the plant.

5. Lines 233 to 235, this sentence implies that these plant proteins/genes are differentially regulated upon infection. This is interesting and it should be also discussed. Is there any other evidence of this? At the gene level? During other interactions. Please discuss.

We have shown that the expression of the two maize genes GRMZM2G043878 (termed AFP1) and GRMZM2G334181 (termed AFP2) is induced during the *U. maydis* infection (Supplementary Fig. 9). It has also been reported that the expression of secreted DUF26 domain proteins in rice is induced upon *Magnaporthe oryzae* infection (Kim et al 2013 *J. Proteomics* 78:58-71). Therefore we consider the induction of secreted DUF26 domain proteins to against fungal invaders to be a general defense response. This has been made clearer in the text and the Kim et al. reference is now included.

In this context we have also shown that plants silenced for AFP1/AFP2 become more susceptible to both *U. maydis* and *C. graminicola* suggesting that the antifungal activity of AFP1 or other secreted DUF26-domain proteins is directed against several fungal pathogens of maize. We have, however, not shown yet which DUF26 proteins are induced after maize colonization by *C. graminicola*.

6. Line 244, “AFP1-His could pull down Rsp3-HA, chimeric Rsp3Um-Sr-HA and Rsp39CA-HA (Fig. 5a-b). However, Rsp3Δ412-869-HA poorly interacted with AFP1-His (Fig. 5a), suggesting that the interaction with AFP1 is largely occurring through the C-terminal repetitive domain of Rsp3.” I do not see this clearly in the corresponding Figure, the Rsp39CA-HA is almost as poorly pulled down Rsp3Δ412-869-HA. I think this figure clearly demonstrates that Rsp3 interacts with AFP1, but I do not agree with the conclusion that the carboxy terminus containing the repeats is the only region responsible for the binding, because loss of the 9 Cysteins (why was not the version without the whole Cys domain used?) also poorly pulls down Rsp3. This also makes sense in terms of virulence, because both strains are the less virulent than the wild type, supporting the hypothesis that interaction with AFC1 is critical for virulence. See below.

To make this more convincing, we have redone the pull-down experiment this time starting with similar amounts of input proteins for comparison. This now more clearly shows that AFP1-His can interact comparably well with Rsp3 and Rsp3_{9CA} while Rsp3Δ₄₁₂₋₈₆₉ still poorly interacted with AFP1-His. We have substituted Fig. 5a with this new figure.

We decided to use Rsp3_{9CA} in the binding experiments rather than the deletion variant because in comparison with wild type Rsp3 this version has more subtle changes than the deletion variant which is lacking about 90 amino acids.

7. Line 252, “Rsp3Δ412-869-HA was only partially able to complement the virulence phenotype of an *rsp3* deletion strain (Supplementary Fig. 4d).” To me, it does not complement at all if compared with the phenotype of the Δ*rsp3* shown in Figure 2. If analysed together, it can be observed that there are major differences in the number of heavy tumors and normal tumors between the two experiments employing the Δ*rsp3* strain shown in Figure 2a and Supp. 4d, being the virulence phenotype of the deletion stronger in Supp. 4d, and thus, the differences to the other strains, such as Rsp3Δ412-869-HA, more exacerbated. Please show all those plant phenotypes together to be able to compare. If this is the case, the strength of the conclusions should be toned down.

We have not done all infections simultaneously, because this would blow up our glass house capacity. However, we now provide more detailed information in Methods how the infection experiments are done and which data sets can be directly compared. There it now reads:

Plant infections were performed in a glasshouse as described and disease severity was scored at 12 dpi following the disease rating established previously². In general, about 40 plants were infected by the same strain in a single experiment and results from three independent infections done in the same season were combined in the respective graphs depicted. Because of seasonal differences in amounts of additional sunlight, disease severity can vary in experiments done at different times in the year. To take this into account the wild type SG200 and the *rsp3* mutant were included in all assays to allow direct comparisons with strains to be tested for virulence. As a consequence, the virulence of strains tested in different experiments,

i.e. Figures, cannot be directly compared and has to be assessed in relation to the control strains tested at the same time.

8. Line 296, “This illustrates that AFP1 has antifungal activity and mannose binding is crucial for this activity.” As mentioned in my previous review, this statement is not fully sustained without further proof, like the titration of the binding by adding mannose that should outcompete the binding to *rsp3* and thus the antifungal activity. If no other data are supplied, I understand the other difficulties the authors mention, the statements again should be moderated.

We have down-toned all experiments concerning the antifungal activity. In addition to showing two technical repeats we now also include quantitative data from three biological repeats and indicate statistically significant differences.

Reviewer #2 (Remarks to the Author):

The revised version of the manuscript by Ma et al., several critical points raised by the reviewers have been addressed. No need to highlight again the wealth of data and the novelty of the finding that *Rsp3* allows the pathogen escaping the activity of AFP1/2. Among the improvements were the inclusion of another, rather distantly related, plant pathogenic fungus, *C. graminicola*. This added to understanding the specificity of the *Ustilago* effector *Rsp3*, and of the maize defense proteins AFP1 and 2. Unfortunately, the glycan binding specificity of *Rsp3* has been addressed by probing a RayBiotech-300-compounds array. This approach led to inconclusive results, and a 600-compounds-array is under way. I found this irritating and way wondering why the authors did not have the patience to wait for the answer. I understand that careful binding assays might take a lot of effort. However, as the binding specificity of the effector is important in order to understand its mode of protection, one may not share the author's view that this is out of scope of this manuscript. This still is a weak point of the manuscript.

We have finally obtained results from the glycan array data with 600 compounds after waiting for a long time (more than the 3 month allowed for the first revision). In this experiment, *Rsp3*-HA does not show binding to chitin substrates, but rather shows highest binding to a blood group A related structure # 389 [GalNAca1-3(Fuca1-2)Galb1-3GalNAca1-3(Fuca1-2)Galb1-4GlcNAcb-Sp0; RFU=32129]. However, by testing the anti-HA antibody alone which was used to detect the *Rsp3*-HA, a very similar profile was observed albeit at a lower RFU (RFU=2195). At present we do not know how to interpret the data and scientists at CFG who conducted the experiment consider the results inconclusive.

In the meantime we have also used the RayBiotech-300-compounds array again and have performed an antibody-alone incubation. The resulting data suggest that *Rsp3*-HA binding to aminoglycoside structures might be specific (Glycan #96-98; *Rsp3*_RFU= 1011, 5910 and 2796 vs antibody alone RFU=334, 1761, 220, respectively). If correct, the binding to aminoglycosides might hint that *Rsp3* could bind to glucosamine. We have therefore also tested binding to chitin and chitosan, cellulose and xylan. Depending on buffer conditions and pH we could observe binding to chitin and chitosan. However, *Rsp3* did not bind chitin present on the 600 compound array.

This shows that much more effort is needed before a conclusion on binding specificity and biological relevance can be drawn. We have also discussed our results with several experts from the field who all agree that the present data is inconclusive. Therefore, we have to stick to our view that elucidating the binding specificity of Rsp3 is beyond the scope of this manuscript.

Further regarding the specificity of the effector, the authors tested binding of Rsp3 to *S. cerevisiae* and *C. graminicola* cells. Given the enormous variability of carbohydrate exposition on hyphae at different stages of infection, one may question whether it is sufficient to look at binding of Rsp3 to conidia. The expression data shown indicate that the Rsp3 gene is most strongly expressed at the initial biotrophic stage. So, how does this relate to *C. graminicola* conidia or yeast cells?

We cannot comment on the last question because yeast does not infect plants. However, in the meantime we did test the binding of Rsp3-HA to *C. graminicola* filaments. We observed similar fluorescence signals in the samples with or without adding Rsp3-HA (see below). Some intensive signals (spots) detected on the hyphae incubated with Rsp3-HA did not show a specific binding pattern, and we consider it likely that these are signals from aggregated Rsp3-HA proteins.

Whether these *C. graminicola* hyphae generated in culture have a surface which is comparable to the surface when *C. graminicola* grows biotrophically is unclear. We have therefore decided not to include this Figure in the supplement. Because of the absence of binding to filaments we cannot comment on how this related to *C. graminicola* conidia or yeast cells. We have made it clear in the text that failure to detect binding to *C. graminicola* conidia or budding cells of *S. cerevisiae* does not exclude binding but may rather reflect that the substrate for binding may not be accessible in these growth stages.

C. graminicola

C. graminicola
+ Rsp3-HA

So, while several critical points have been taken care of, some questions remain to be answered.

Specific comments:

Lines 104 and 106: DNA sizes should be given in a consistent way, either bp or kbp.

This has been changed now to kbp throughout the manuscript

Line 217: ... also when added externally.

This has been changed

Lines 221 - 223: Comments on the binding specificity of Rsp3 are to be found above. Again, as hyphal surfaces may differ, depending on their lifestyle, and certainly differ from spore surfaces, data shown in Suppl. Fig. 6a and b may be difficult to interpret.

We fully agree with this reviewer and have been very careful in interpreting our results (see our more detailed comment on previous page).

Line 232: Sentence is incomplete.

Sentence has been changed.

Line 238: The abbreviation AFP should be explained at first mention.

This is now explained at first mention.

Lines 295 and 296, and Fig. 6c: The authors estimate titers (about 4 to 5-fold lower...). Can more precise data be provided?

We have now added new sub-figures giving quantitative results from three biological replicates (Figure 6d) and by indicating significant differences.

Lines 302 - 304: This sentence gives redundant information (see lines 295 and 296) and should be deleted.

We do not consider these lines to give redundant information. If this reviewer considers 295-296 and 289-290 to be redundant we have to point out that in 295-296 binding of Rsp3 is shown to protect against AFP1 while in 289-290 it is shown that AFP1 has anti-fungal activity while the AFP1** protein has not.

Fig. 7 shows infection assays. Again, standard deviations are missing in the *Ustilago* assays. The differences of *Colletotrichum* disease symptoms on FoMV as compared to FoMV_AFP1/2 leaves are obvious. I strongly suggest to use qPCR to quantify fungal DNA, and to use that as a quantitative criterion of fungal pathogenic development. Counting spots is rarely reproducible, which may be the reason why standard deviations are missing in Fig. 7d as well. As mannose-binding AFP1/2 affect the *C. graminicola* infection process and disease severity, data shown in Suppl. Fig. 6a may even be harder to interpret. Labeling cross-sectioned biotrophic and necrotrophic hyphae may lead to improved understanding of the specificity of Rsp3 and AFP1/2. Furthermore, as Rsp3 appears to play a prominent role at early biotrophic stages of infection, would the infection process of a necrotroph be altered in virulence in the presence/absence of AFP1/2?

In experiments in Fig. 7b and Fig. 7d, we show two independent biological repeats to make this data comparable to the infections with the *rsp3* mutant in Fig. 7a (where we analyze silencing efficiency by qRT-PCR in every single plant). We feel that showing the results from two independent experiments is sufficient to make the point that silencing rescues the virulence defect of the *rsp3* mutant and increases the virulence of wild type *U. maydis* as well as virulence of *C. graminicola*.

In Fig 7c-d, the disease symptoms of *C. graminicola* are very obvious, and were quantified by counting the spots in two independent experiments which are both shown. In our hands the symptoms are reproducible and we therefore see no need to determine fungal biomass. We have discussed this with two experts in *C. graminicola* who found the data entirely convincing.

Of course, it would now also be possible to test whether AFPs affect virulence necrotrophic fungi, but having shown the effects on *C. graminicola* we feel that this already allows us to generalize and conclude that AFPs do not only have antifungal activity against *U. maydis*.

We cannot follow the comment by this referee that labeling of cross-sectioned necrotrophic and biotrophic hyphae may lead to an improved understanding of the specificity of Rsp3 and AFP1/2 because neither *C. graminicola* nor *B. cinerea* encode *rsp3*-related genes and at this stage it is not clear whether AFP1/2 or additional DUF26 proteins are differentially expressed in these interactions.

REVIEWERS' COMMENTS:

Reviewer #1 (Remarks to the Author):

The authors have addressed all issues raised by the both referees, added several new data and incorporated some of the corrections. I am satisfied with all of them, except with the reply to my previous point 7. Although, I fully understand that not all experiments can be carried out simultaneously, it is somehow misleading to say that they can not be compared from Figure to Figure because then the meaning of a given result might not be valid in a future repetition of the experiment. Furthermore, small differences such as "only partially able to complement" might not be observable in a new set of experiments. I think it would be good to somehow take into consideration that problem and rather say that that strain poorly complemented if at all.

Reviewer #2 (Remarks to the Author):

Ma et al. have resubmitted a second revision of their above manuscript on the Rsp3 effector of the maize smut fungus. In this manuscript most of the previous concerns have been taken care of. The authors show convincingly that although Rsp3 is not a pathogenicity determinant, it is an important virulence factor of the biotroph *Ustilago maydis*. The effector is a secreted protein decorating the invading hyphae.

Regarding the binding sites of the effector on the hyphal surface, the manuscript did not progress significantly. The results are still inconclusive, as stated by the authors in their response letter. This leaves the manuscript - and the model shown in Fig. 8 - with an open question. Thus, this weak point of the manuscript still exists. However, this may be acceptable, as this is not the central point the manuscript makes. In their response letter the authors further comment on the binding patterns of Rsp3-HA on yeast cells of *U. maydis* and on filaments of *C. graminicola*. The filaments used were vegetative hyphae. Not only in *C. graminicola*, but also in other hemibiotrophs such as *Magnaporthe oryzae* and *Zymoseptoria tritici* cell wall modifications have been shown to occur during the invasion process, and it may not be surprising that labeling experiments with non-pathogenic hyphae. The authors may want to comment on this in more detail.

Including new Fig. 6d strengthens the manuscript and is appreciated.

Fig. 7 still needs some attention. While standard deviations have been added to the quantitative data describing symptom severities, this has still not been done in Figs. 7a, b, and d. Given the enormous effort it takes to do repeat silencing experiments, using the Foxtail mosaic virus system, I do understand that the authors resist to do qPCR experiments. This is plausible, as the differences in symptom severities are very clear. Still qPCR experiments and symptom assessment score distinct parameters. One may compromise on this point, although this may be regarded as a weakness of the manuscript.

Reviewer #1 (Remarks to the Author):

The authors have addressed all issues raised by the both referees, added several new data and incorporated some of the corrections. I am satisfied with all of them, except with the reply to my previous point 7. Although, I fully understand that not all experiments can be carried out simultaneously, it is somehow misleading to say that they can not be compared from Figure to Figure because then the meaning of a given result might not be valid in a future repetition of the experiment. Furthermore, small differences such as "only partially able to complement" might not be observable in a new set of experiments. I think it would be good to somehow take into consideration that problem and rather say that that strain poorly complemented if at all.

We have now written that the strain poorly complemented. However, we have not added "if at all" because we have included in Supplementary Fig.4b the wild type strain, the deletion mutant and the strain to be tested. This experiment was done in three biological replicates and each time the mutant protein poorly complemented. This documents that the phenotype "poor complementation" is reproducible. What the reviewer is doing, namely comparing the disease symptoms of individual strains from different experiments without taking into account the respective disease scores of wild type and mutant strains conducted at the same time is in our view not allowed. What this reviewer in fact is asking for is a normalization of strain phenotypes with respect to wild type. This could be done in principle, however, we feel that our representation of the data is more honest and fully accepted in the smut community.

Reviewer #2 (Remarks to the Author):

Ma et al. have resubmitted a second revision of their above manuscript on the Rsp3 effector of the maize smut fungus. In this manuscript most of the previous concerns have been taken care of. The authors show convincingly that although Rsp3 is not a pathogenicity determinant, it is an important virulence factor of the biotroph *Ustilago maydis*. The effector is a secreted protein decorating the invading hyphae.

Regarding the binding sites of the effector on the hyphal surface, the manuscript did not progress significantly. The results are still inconclusive, as stated by the authors in their response letter. This leaves the manuscript - and the model shown in Fig. 8 - with an open question. Thus, this weak point of the manuscript still exists. However, this may be acceptable, as this is not the central point the manuscript makes. In their response letter the authors further comment on the binding patterns of Rsp3-HA on yeast cells of *U. maydis* and on filaments of *C. graminicola*. The filaments used were vegetative hyphae. Not only in *C. graminicola*, but also in other hemibiotrophs such as *Magnaporthe oryzae* and *Zymoseptoria tritici* cell wall modifications have been shown to occur during the invasion process, and it may not be surprising that labeling experiments with non-pathogenic hyphae. The authors may want to comment on this in more detail.

We have now extended the main text taking into consideration these comments and have also added a recent review as reference.

Including new Fig. 6d strengthens the manuscript and is appreciated.

Fig. 7 still needs some attention. While standard deviations have been added to the quantitative data describing symptom severities, this has still not been done in Figs. 7a, b, and d. Given the enormous effort it takes to do repeat silencing experiments, using the Foxtail mosaic virus system, I do understand that the authors resist to do qPCR experiments. This is plausible, as the differences in symptom severities are very clear. Still qPCR experiments and symptom assessment score distinct parameters. One may compromise on this point, although this may be regarded as a weakness of the manuscript.

We are aware of this weakness, which did arise because in the initial FoMV silencing experiments and the replicate we assessed each individual plant (70 plants in total) for the silencing of AFP1 and AFP2 (Fig.7a and Supplementary Fig.10). By doing so, we realized that in some plants which were silenced for AFP1 and AFP2, AFP2 gene transcription was upregulated. Rather than assessing virulence only in the plants which were silenced for both genes we decided to take all, also to avoid that we would have to assess silencing in each individual plant in all subsequent experiments. This means that in all likelihood the differences in virulence in control plants and plants truly silenced in both genes might even be more prominent. We have decided to present the two biological replicates of the initial silencing experiment separately, because we cannot calculate the standard deviation from two experiments and feel that the presentation of both biological replicates allows a fair comparison and assessment. We have then followed this scheme also in the subsequent experiments where we have assessed virulence of *U. maydis* (Fig.7b) and *C. graminicola* (Fig.7c and d) macroscopically. We doubt that we would gain more information from determining fungal biomass by qPCR, although the reviewer is of course right that biomass and macroscopic symptoms measure two different parameters. However, we have throughout the manuscript used macroscopic symptoms also to assess disease of *U. maydis*, and this is fully accepted in a system where disease severity can be scored macroscopically in leaves. We agree that a symptomless infection needs to be followed by qPCR. As we can score and quantify macroscopic symptoms for *C. graminicola* also in leaves we relied to these macroscopic symptoms to assess disease severity also here.